# Online Robust Reinforcement Learning with Model Uncertainty

**Yue Wang**
University at Buffalo
Buffalo, NY 14228
ywang294@buffalo.edu

**Shaofeng Zou**
University at Buffalo
Buffalo, NY 14228
szou3@buffalo.edu

## Abstract

Robust reinforcement learning (RL) is to find a policy that optimizes the worst-case performance over an uncertainty set of MDPs. In this paper, we focus on *model-free* robust RL, where the uncertainty set is defined to be centering at a misspecified MDP that generates a single sample trajectory sequentially, and is assumed to be *unknown*. We develop a sample-based approach to estimate the unknown uncertainty set, and design robust Q-learning algorithm (tabular case) and robust TDC algorithm (function approximation setting), which can be implemented in an online and incremental fashion. For the robust Q-learning algorithm, we prove that it converges to the optimal robust Q function, and for the robust TDC algorithm, we prove that it converges asymptotically to some stationary points. Unlike the results in [Roy et al., 2017], our algorithms do not need any additional conditions on the discount factor to guarantee the convergence. We further characterize the finite-time error bounds of the two algorithms, and show that both the robust Q-learning and robust TDC algorithms converge as fast as their vanilla counterparts (within a constant factor). Our numerical experiments further demonstrate the robustness of our algorithms. Our approach can be readily extended to robustify many other algorithms, e.g., TD, SARSA, and other GTD algorithms.

## 1   Introduction

Existing studies on Markov decision process (MDP) and reinforcement learning (RL) [Sutton and Barto, 2018] mostly rely on the crucial assumption that the environment on which a learned policy will be deployed is the same one that was used to generate the policy, which is often violated in practice – e.g., the simulator may be different from the true environment, and the MDP may evolve over time. Due to such model deviation, the actual performance of the learned policy can significantly degrade. To address this problem, the framework of robust MDP was formulated in [Bagnell et al., 2001, Nilim and El Ghaoui, 2004, Iyengar, 2005], where the transition kernel of the MDP is not fixed and lies in an uncertainty set, and the goal is to learn a policy that performs well under the worst-case MDP in the uncertainty set. In [Bagnell et al., 2001, Nilim and El Ghaoui, 2004, Iyengar, 2005], it was assumed that the uncertainty set is known beforehand, i.e., model-based approach, and dynamic programming can be used to find the optimal robust policy.

The model-based approach, however, requires a model of the uncertainty set known beforehand, and needs a large memory to store the model when the state and action spaces are large, which make it less applicable for many practical scenarios. This motivates the study in this paper, *model-free* robust RL with model uncertainty, which is to learn a robust policy using a single sample trajectory from a misspecified MDP, e.g., a simulator and a similar environment in which samples are easier to collect than in the target environment where the policy is going to be deployed. The major challenge lies in that the transition kernel of the misspecified MDP is not given beforehand, and thus, the uncertainty

35th Conference on Neural Information Processing Systems (NeurIPS 2021).

set and the optimal robust policy need to be learned simultaneously using sequentially observed data from the misspecified MDP. Moreover, robust RL learns the value function of the worst-case MDP in the uncertainty set which is different from the misspecified MDP that generates samples. This is similar to the off-policy learning, which we refer to as the "off-transition-kernel" setting. Therefore, the learning may be unstable and could diverge especially when function approximation is used [Baird, 1995].

In this paper, we develop a model-free approach for robust RL with model uncertainty. Our major contributions in this paper are summarized as follows.

- Motivated by empirical studies of adversarial training in RL [Huang et al., 2017, Kos and Song, 2017, Lin et al., 2017, Pattanaik et al., 2018, Mandlekar et al., 2017] and the $R$-contamination model in robust detection (called $\epsilon$-contamination model in [Huber, 1965]), we design the uncertainty set using the $R$-contamination model (see (4) for the details). We then develop an approach to estimate the unknown uncertainty set using only the current sample, which does not incur any additional memory cost. Unlike the approach in [Roy et al., 2017], where the uncertainty set is relaxed to one not depending on the misspecified MDP that generates samples so that an online algorithm can be constructed, our approach does not need to relax the uncertainty set.

- We develop a robust Q-learning algorithm for the tabular case, which can be implemented in an online and incremental fashion, and has the same memory cost as the vanilla Q-learning algorithm. We show that our robust Q-learning algorithm converges asymptotically, and further characterize its finite-time error bound. Unlike the results in [Roy et al., 2017] where a stringent condition on the discount factor (which is due to the relaxation of the uncertainty set, and prevents the use of a discount factor close to 1 in practice) is needed to guarantee the convergence, our algorithm converges without the need of such condition. Furthermore, our robust Q-learning algorithm converges as fast as the vanilla Q-learning algorithm [Li et al., 2020] (within a constant factor), while being robust to model uncertainty.

- We generalize our approach to the case with function approximation (for large state/action space). We investigate the robust policy evaluation problem, i.e., evaluate a given policy under the worst-case MDP in the uncertainty set. As mentioned before, the robust RL problem is essentially "off-transition-kernel", and therefore non-robust methods with function approximation may diverge [Baird, 1995] (also see our experiments). We develop a novel extension of the gradient TD (GTD) method [Maei et al., 2010, Maei, 2011, Sutton et al., 2008] to robust RL. Our approach introduces a novel smoothed robust Bellman operator to construct the smoothed mean-squared projected robust Bellman error (MSPRBE). Using our uncertainty set design and online sample-based estimation, we develop a two time-scale robust TDC algorithm. We further characterize its convergence and finite-time error bound.

- We conduct numerical experiments to validate the robustness of our approach. In our experiments, our robust Q-learning algorithm achieves a much higher reward than the vanilla Q-learning algorithm when being trained on a misspecified MDP; and our robust TDC algorithm converges much faster than the vanilla TDC algorithm, and the vanilla TDC algorithm may even diverge.

## 1.1 Related Work

**Model-Based Robust MDP.** The framework of robust MDP was investigated in [Iyengar, 2005, Nilim and El Ghaoui, 2004, Bagnell et al., 2001, Satia and Lave Jr, 1973, Wiesemann et al., 2013], where the transition kernel is assumed to be in some uncertainty set, and the problem can be solved by dynamic programming. This approach was further extended to the case with function approximation in [Tamar et al., 2014]. However, these studies are model-based, which assume beforehand knowledge of the uncertainty set. In this paper, we investigate the model-free setting, where the uncertainty set is a set of MDPs centered around some unknown Markov transition kernel from which a single sample trajectory can be sequentially observed.

**Adversarial Robust RL.** It was shown in [Iyengar, 2005] that the robust MDP problem is equivalent to a zero-sum game between the agent and the nature. Motivated by this fact, the adversarial training approach, where an adversary perturbs the state transition, was studied in [Vinitsky et al., 2020, Pinto et al., 2017, Abdullah et al., 2019, Hou et al., 2020, Rajeswaran et al., 2017, Atkeson and Morimoto, 2003, Morimoto and Doya, 2005]. This method relies on a simulator, where the state transition can be modified in an arbitrary way. Another approach is to modify the current state through adversarial

samples, which is more heuristic, e.g., [Huang et al., 2017, Kos and Song, 2017, Lin et al., 2017, Pattanaik et al., 2018, Mandlekar et al., 2017]. Despite the empirical success of these approaches, theoretical performance guarantees, e.g., convergence to the optimal robust policy and convergence rate, are yet to be established. The main difference lies in that during the training, our approach does not need to manipulate the state transition of the MDP. More importantly, we develop the asymptotic convergence to the optimal robust policy and further characterize the finite-time error bound. In [Lim et al., 2013], the scenario where some unknown parts of the state space can have arbitrary transitions while other parts are purely stochastic was studied. Adaptive algorithm to adversarial behavior was designed, and its regret bound is shown to be similar to the purely stochastic case. In [Zhang et al., 2020a], the robust adversarial RL problem for the special linear quadratic case was investigated.

**Model-free Robust RL.** In [Roy et al., 2017, Badrinath and Kalathil, 2021] model-free RL with model uncertainty was studied, where in order to construct an algorithm that can be implemented in an online and incremental fashion, the uncertainty set was firstly relaxed by dropping the dependency on the misspecified MDP that generates the samples (centroid of the uncertainty set). Such a relaxation is pessimistic since the relaxed uncertainty set is not centered at the misspecified MDP anymore (which is usually similar to the target MDP), making the robustness to the relaxed uncertainty set not well-justified. Such a relaxation will further incur a stringent condition on the discounted factor to guarantee the convergence, which prevents the use of a discount factor close to 1 in practice. Moreover, only asymptotic convergence was established in [Roy et al., 2017]. In this paper, we do not relax the uncertainty set, and instead propose an online approach to estimate it. Our algorithms converge without the need of the condition on the discount factor. We also provides finite-time error bounds for our algorithms. The multi-agent RL robust to reward uncertainty was investigated in [Zhang et al., 2020b], where the reward uncertainty set is known, but the transition kernel is fixed.

**Finite-time Error Bound for RL Algorithms.** For the tubular case, Q-learning has been studied intensively, e.g., in [Even-Dar et al., 2003, Beck and Srikant, 2012, Qu and Wierman, 2020, Li et al., 2020, Wainwright, 2019, Li et al., 2021]. TD with function approximation were studied in [Dalal, Gal and Szörényi, Balázs and Thoppe, Gugan and Mannor, Shie, 2018, Bhandari et al., 2018, Srikant and Ying, 2019, Cai et al., 2019, Sun et al., 2020]. Q-learning and SARSA with linear function approximation were investigated in [Zou et al., 2019, Chen et al., 2019]. The finite-time error bounds for the gradient TD algorithms [Maei et al., 2010, Sutton et al., 2009, Maei et al., 2010] were further developed recently in [Dalal et al., 2018, Liu et al., 2015, Gupta et al., 2019, Xu et al., 2019, Dalal et al., 2020, Kaledin et al., 2020, Ma et al., 2020, Wang and Zou, 2020, Ma et al., 2021, Doan, 2021]. There are also finite-time error bounds on the policy gradient methods and actor critic methods, e.g., [Wang et al., 2020, Yang et al., 2019, Kumar et al., 2019, Qiu et al., 2019, Wu et al., 2020, Cen et al., 2020, Bhandari and Russo, 2019, Agarwal et al., 2021, Mei et al., 2020]. We note that these studies are for the non-robust RL algorithms, and in this paper, we design robust RL algorithms, and characterize their finite-time error bounds.

## 2 Preliminaries

**Markov Decision Process.** An MDP can be characterized by a tuple $(\mathcal{S}, \mathcal{A}, \mathsf{P}, c, \gamma)$, where $\mathcal{S}$ and $\mathcal{A}$ are the state and action spaces, $\mathsf{P} = \left\{ p_s^a \in \Delta_{|\mathcal{S}|}, a \in \mathcal{A}, s \in \mathcal{S} \right\}$ is the transition kernel[1], $c$ is the cost function, and $\gamma \in [0, 1)$ is the discount factor. Specifically, $p_s^a$ denotes the distribution of the next state if taking action $a$ at state $s$. Let $p_s^a = \{p_{s,s'}^a\}_{s' \in \mathcal{S}}$, where $p_{s,s'}^a$ denotes the probability that the environment transits to state $s'$ if taking action $a$ at state $s$. The cost of taking action $a$ at state $s$ is given by $c(s, a)$. A stationary policy $\pi$ is a mapping from $\mathcal{S}$ to a distribution over $\mathcal{A}$. At each time $t$, an agent takes an action $A_t \in \mathcal{A}$ at state $S_t \in \mathcal{S}$. The environment then transits to the next state $S_{t+1}$ with probability $p_{S_t, S_{t+1}}^{A_t}$, and the agent receives cost given by $c(S_t, A_t)$. The value function of a policy $\pi$ starting from any initial state $s \in \mathcal{S}$ is defined as the expected accumulated discounted cost by following $\pi$: $\mathbb{E}\left[\sum_{t=0}^{\infty} \gamma^t c(S_t, A_t) | S_0 = s, \pi\right]$, and the goal is to find the policy $\pi$ that minimizes the above value function for any initial state $s \in \mathcal{S}$.

**Robust Markov Decision Process.** In the robust case, the transition kernel is not fixed and lies in some uncertainty set. Denote the transition kernel at time $t$ by $\mathsf{P}_t$, and let $\kappa = (\mathsf{P}_0, \mathsf{P}_1, ...)$, where $\mathsf{P}_t \in \mathbf{P}, \forall t \geq 0$, and $\mathbf{P}$ is the uncertainty set of the transition kernel. The sequence $\kappa$ can be viewed as the policy of the nature, and is adversarially chosen by the nature [Bagnell et al., 2001, Nilim and

---

[1] $\Delta_n$ denotes the $(n-1)$-dimensional probability simplex: $\{(p_1, ..., p_n) | 0 \leq p_i \leq 1, \sum_{i=1}^{n} p_i = 1\}$.

El Ghaoui, 2004, Iyengar, 2005]. Define the robust value function of a policy $\pi$ as the worst-case expected accumulated discounted cost following a fixed policy $\pi$ over all transition kernels in the uncertainty set:

$$V^\pi(s) = \max_\kappa \mathbb{E}_\kappa \left[ \sum_{t=0}^\infty \gamma^t c(S_t, A_t) | S_0 = s, \pi \right], \tag{1}$$

where $\mathbb{E}_\kappa$ denotes the expectation when the state transits according to $\kappa$. Similarly, define the robust action-value function for a policy $\pi$: $Q^\pi(s, a) = \max_\kappa \mathbb{E}_\kappa \left[ \sum_{t=0}^\infty \gamma^t c(S_t, A_t) | S_0 = s, A_0 = a, \pi \right]$. The goal of robust RL is to find the optimal robust policy $\pi^*$ that minimizes the worst-case accumulated discounted cost:

$$\pi^* = \arg\min_\pi V^\pi(s), \forall s \in \mathcal{S}. \tag{2}$$

We also denote $V^{\pi^*}$ and $Q^{\pi^*}$ by $V^*$ and $Q^*$, respectively, and $V^*(s) = \min_{a \in \mathcal{A}} Q^*(s, a)$.

Note that a transition kernel is a collection of conditional distributions. Therefore, the uncertainty set $\mathbf{P}$ of the transition kernel can be equivalently written as a collection of $\mathcal{P}_s^a$ for all $s \in \mathcal{S}, a \in \mathcal{A}$, where $\mathcal{P}_s^a$ is a set of conditional distributions $p_s^a$ over the state space $\mathcal{S}$. Denote by $\sigma_\mathcal{P}(v) \triangleq \max_{p \in \mathcal{P}} (p^\top v)$ the support function of vector $v$ over a set of probability distributions $\mathcal{P}$. For robust MDP, the following robust analogue of the Bellman recursion was provided in [Nilim and El Ghaoui, 2004, Iyengar, 2005].

**Theorem 1.** *[Nilim and El Ghaoui, 2004] The following perfect duality condition holds for all $s \in \mathcal{S}$:*

$$\min_\pi \max_\kappa \mathbb{E}_\kappa \left[ \sum_{t=0}^\infty \gamma^t c(S_t, A_t) \big| \pi, S_0 = s \right] = \max_\kappa \min_\pi \mathbb{E}_\kappa \left[ \sum_{t=0}^\infty \gamma^t c(S_t, A_t) \big| \pi, S_0 = s \right]. \tag{3}$$

*The optimal robust value function $V^*$ satisfies $V^*(s) = \min_{a \in \mathcal{A}} (c(s, a) + \gamma \sigma_{\mathcal{P}_s^a}(V^*))$, and the optimal robust action-value function $Q^*$ satisfies $Q^*(s, a) = c(s, a) + \gamma \sigma_{\mathcal{P}_s^a}(V^*)$.*

Define the robust Bellman operator $\mathbf{T}$ by $\mathbf{T}Q(s, a) = c(s, a) + \gamma \sigma_{\mathcal{P}_s^a}(\min_{a \in \mathcal{A}} Q(s, a))$. It was shown in [Nilim and El Ghaoui, 2004, Iyengar, 2005] that $\mathbf{T}$ is a contraction and its fixed point is the optimal robust $Q^*$. When the uncertainty set is known, so that $\sigma_{\mathcal{P}_s^a}$ can be computed exactly, $V^*$ and $Q^*$ can be solved by dynamic programming [Iyengar, 2005, Nilim and El Ghaoui, 2004].

## 3 R-Contamination Model For Uncertainty Set Construction

In this section, we construct the uncertainty set using the $R$-contamination model.

Let $\mathsf{P} = \{p_s^a, s \in \mathcal{S}, a \in \mathcal{A}\}$ be the centroid of the uncertainty set, i.e., the transition kernel that generates the sample trajectory, and $\mathsf{P}$ is *unknown*. For example, $\mathsf{P}$ can be the simulator at hand, which may not be exactly accurate; and $\mathsf{P}$ can be the transition kernel of environment 1, from which we can take samples to learn a policy that will be deployed in a similar environment 2. The goal is to learn a policy using samples from $\mathsf{P}$ that performs well when applied to a perturbed MDP from $\mathsf{P}$.

Motivated by empirical studies of adversarial training in RL [Huang et al., 2017, Kos and Song, 2017, Lin et al., 2017, Pattanaik et al., 2018, Mandlekar et al., 2017] and the $R$-contamination model in robust detection [Huber, 1965], we use the $R$-contamination model to define the uncertainty set:

$$\mathcal{P}_s^a = \left\{ (1-R)p_s^a + Rq | q \in \Delta_{|\mathcal{S}|} \right\}, s \in \mathcal{S}, a \in \mathcal{A}, \text{ for some } 0 \le R \le 1. \tag{4}$$

Here, $p_s^a$ is the centroid of the uncertainty set $\mathcal{P}_s^a$ at $(s, a)$, which is *unknown*, and $R$ is the design parameter of the uncertainty set, which measures the size of the uncertainty set, and is assumed to be known in the algorithm. We then let $\mathbf{P} = \bigotimes_{s \in \mathcal{S}, a \in \mathcal{A}} \mathcal{P}_s^a$.

**Remark 1.** *$R$-contamination model is closely related to other uncertainty set models like total variation and KL-divergence. It can be shown that $R$-contamination set certered at $p$ is a subset of total variation ball : $\left\{ (1-R)p + Rq | q \in \Delta_{|\mathcal{S}|} \right\} \subset \left\{ q \in \Delta_{|\mathcal{S}|} | d_{TV}(p, q) \le R \right\}$. Hence the total variation uncertainty set is less conservative than our $R$-contamination uncertainty set. KL-divergence moreover can be related to total variation using Pinsker's inequality, i.e., $d_{TV}(p, q) \le \sqrt{\frac{1}{2} d_{KL}(p, q)}$.*

# 4 Tabular Case: Robust Q-Learning

In this section, we focus on the tabular case with finite state and action spaces. We focus on the asynchronous setting where a single sample trajectory is available with Markovian noise. We will develop an efficient approach to estimate the unknown uncertainty set $\mathbf{P}$, and further the support function $\sigma_{\mathcal{P}_s^a}(\cdot)$, and then design our robust Q-learning algorithm.

We propose an efficient and data-driven approach to estimate the unknown $p_s^a$ and thus the unknown uncertainty set $\mathcal{P}_s^a$ for any $s \in \mathcal{S}$ and $a \in \mathcal{A}$. Specifically, denote the sample at $t$-th time step by $O_t = (s_t, a_t, s_{t+1})$. We then use $O_t$ to obtain the maximum likelihood estimate (MLE) $\hat{p}_t \triangleq \mathbb{1}_{s_{t+1}}$ of the transition kernel $p_{s_t}^{a_t}$, where $\mathbb{1}_{s_{t+1}}$ is a probability distribution taking probability 1 at $s_{t+1}$ and 0 at other states. This is an unbiased estimate of the transition kernel $p_{s_t}^{a_t}$ conditioning on $S_t = s_t$ and $A_t = a_t$. We then design a sample-based estimate $\hat{\mathcal{P}}_t \triangleq \{(1-R)\hat{p}_t + Rq | q \in \Delta_{|\mathcal{S}|}\}$ of the uncertainty set $\mathcal{P}_{s_t}^{a_t}$. Using the sample-based uncertainty set $\hat{\mathcal{P}}_t$, we construct the following robust Q-learning algorithm in Algorithm 1. For any $t$, $\sigma_{\hat{\mathcal{P}}_t}(V_t)$ can be easily computed: $\sigma_{\hat{\mathcal{P}}_t}(V_t) =$

---

**Algorithm 1** Robust Q-Learning

**Initialization**: $T$, $Q_0(s, a)$ for all $(s, a) \in \mathcal{S} \times \mathcal{A}$, behavior policy $\pi_b$, $s_0$, step size $\alpha_t$
1: **for** $t = 0, 1, 2, ..., T-1$ **do**
2:     Choose $a_t$ according to $\pi_b(\cdot|s_t)$
3:     Observe $s_{t+1}$ and $c_t$
4:     $V_t(s) \leftarrow \min_{a \in \mathcal{A}} Q_t(s, a), \forall s \in \mathcal{S}$
5:     $Q_{t+1}(s_t, a_t) \leftarrow (1-\alpha_t)Q_t(s_t, a_t) + \alpha_t(c_t + \gamma\sigma_{\hat{\mathcal{P}}_t}(V_t))$
6:     $Q_{t+1}(s, a) \leftarrow Q_t(s, a)$ for $(s, a) \neq (s_t, a_t)$
7: **end for**
**Output**: $Q_T$

---

$R \max_{s \in \mathcal{S}} V_t(s) + (1-R)V_t(s_{t+1})$. Hence the update in Algorithm 1 (line 5) can be written as

$$Q_{t+1}(s_t, a_t) \leftarrow (1-\alpha_t)Q_t(s_t, a_t) + \alpha_t(c_t + \gamma R \max_{s \in \mathcal{S}} V_t(s) + \gamma(1-R)V_t(s_{t+1})). \tag{5}$$

Compared to the model-based approach, our approach is model-free. It does not require the prior knowledge of the uncertainty set, i.e., the knowledge of $p_s^a, \forall s \in \mathcal{S}, a \in \mathcal{A}$. Furthermore, the memory requirement of our algorithm is $|\mathcal{S}| \times |\mathcal{A}|$ (used to store the Q-table), and unlike the model-based approach it does not need a table of size $|\mathcal{S}|^2|\mathcal{A}|$ to store $p_s^a, \forall s \in \mathcal{S}, a \in \mathcal{A}$, which could be problematic if the state space is large. Moreover, our algorithm does not involve a relaxation of the uncertainty set like the one in [Roy et al., 2017], which will incur a stringent condition on the discount factor to guarantee the convergence. As will be shown below, the convergence of our Algorithm 1 does not require any condition on the discount factor.

We show in the following theorem that the robust Q-learning algorithm converges asymptotically to the optimal robust action-value function $Q^*$.

**Theorem 2.** *(Asymptotic Convergence) If step sizes $\alpha_t$ satisfy that $\sum_{t=0}^{\infty} \alpha_t = \infty$ and $\sum_{t=0}^{\infty} \alpha_t^2 < \infty$, then $Q_t \rightarrow Q^*$ as $t \rightarrow \infty$ with probability 1.*

To further establish the finite-time error bound for our robust Q-learning algorithm in Algorithm 1, we make the following assumption that is commonly used in the analysis of vanilla Q-learning.

**Assumption 1.** *The Markov chain induced by the behavior policy $\pi_b$ and the transition kernel $p_s^a, \forall s \in \mathcal{S}, a \in \mathcal{A}$ is uniformly ergodic.*

Let $\mu_{\pi_b}$ denote the stationary distribution over $\mathcal{S} \times \mathcal{A}$ induced by $\pi_b$ and $p_s^a, \forall s \in \mathcal{S}, a \in \mathcal{A}$. We then further define $\mu_{\min} = \min_{(s,a) \in \mathcal{S} \times \mathcal{A}} \mu_{\pi_b}(s, a)$. This quantity characterizes how many samples are needed to visit every state-action pair sufficiently often. Define the following mixing time of the induced Markov chain: $t_{\text{mix}} = \min\{t : \max_{s \in \mathcal{S}} d_{\text{TV}}(\mu_\pi, P(s_t = \cdot|s_0 = s)) \leq \frac{1}{4}\}$, where $d_{\text{TV}}$ is the total variation distance.

The following theorem establishes the finite-time error bound of our robust Q-learning algorithm.

**Theorem 3.** *(Finite-Time Error Bound) There exist some positive constants $c_0$ and $c_1$ such that for any $\delta < 1$, any $\epsilon < \frac{1}{1-\gamma}$, any $T$ satisfying*

$$T \geq c_0 \left( \frac{1}{\mu_{min}(1-\gamma)^5 \epsilon^2} + \frac{t_{mix}}{\mu_{min}(1-\gamma)} \right) \log \left( \frac{T|\mathcal{S}||\mathcal{A}|}{\delta} \right) \log \left( \frac{1}{\epsilon(1-\gamma)^2} \right), \qquad (6)$$

*and step size $\alpha_t = \frac{c_1}{\log\left(\frac{T|\mathcal{S}||\mathcal{A}|}{\delta}\right)} \min \left( \frac{1}{t_{mix}}, \frac{\epsilon^2(1-\gamma)^4}{\gamma^2} \right), \forall t \geq 0$ we have with probability at least $1 - 6\delta$, $\|Q_T - Q^*\|_\infty \leq 3\epsilon$.*

From the theorem, we can see that to guarantee an $\epsilon$-accurate estimate, a sample size $\tilde{\mathcal{O}}(\frac{1}{\mu_{\min}(1-\gamma)^5\epsilon^2} + \frac{t_{\min}}{\mu_{\min}(1-\gamma)})$ (up to some logarithmic terms) is needed. This complexity matches with the one for the vanilla Q-learning in [Li et al., 2020] (within a constant factor), while our algorithm also guarantees robustness to MDP model uncertainty. Our algorithm design and analysis can be readily extended to robustify TD and SARSA. The variance-reduction technique [Wainwright, 2019] can also be combined with our robust Q-learning algorithm to further improve the dependency on $(1-\gamma)$.

## 5   Function Approximation: Robust TDC

In this section, we investigate the case where the state and action spaces can be large or even continuous. A popular approach is to approximate the value function using a parameterized function, e.g., linear function and neural network. In this section, we focus on the case with linear function approximation to illustrate the main idea of designing robust RL algorithms. Our approach can be extended to non-linear (smooth) function approximation using techniques in, e.g., [Cai et al., 2019, Bhatnagar et al., 2009, Wai et al., 2019, Wang et al., 2021].

We focus on the problem of robust policy evaluation, i.e., estimate the robust value function $V^\pi$ defined in (1) for a given policy $\pi$ under the worst-case MDP transition kernel in the uncertainty set. Note that for robust RL with model uncertainty, any policy evaluation problem can be viewed as "off-transition-kernel", as it is to evaluate the value function under the worst-case MDP using samples from a different MDP. Since the TD algorithm with function approximation may diverge under off-policy training [Baird, 1995] and importance sampling cannot be applied here due to unknown transition kernel, in this paper we generalize the GTD method [Maei et al., 2010, Maei, 2011] to the robust setting.

Let $\left\{ \phi^{(i)} : \mathcal{S} \to \mathbb{R}, i = 1, \ldots, N \right\}$ be a set of $N$ fixed base functions, where $N \ll |\mathcal{S}||\mathcal{A}|$. In particular, we approximate the robust value function using a linear combination of $\phi^{(i)}$'s: $V_\theta(s) = \sum_{i=1}^N \theta^i \phi_s^{(i)} = \phi_s^\top \theta$, where $\theta \in \mathbb{R}^N$ is the weight vector.

Define the following robust Bellman operator for a given policy $\pi$:

$$\mathbf{T}_\pi V(s) \triangleq \mathbb{E}_{A \sim \pi(\cdot|s)}[c(s,A) + \gamma \sigma_{\mathcal{P}_s^A}(V)]$$

$$= \mathbb{E}_{A \sim \pi(\cdot|s)} \left[ c(s,A) + \gamma(1-R) \sum_{s' \in \mathcal{S}} p_{s,s'}^A V(s') + \gamma R \max_{s' \in \mathcal{S}} V(s') \right]. \qquad (7)$$

We then define the mean squared projected robust Bellman error (MSPRBE) as

$$\text{MSPRBE}(\theta) = \|\mathbf{\Pi}\mathbf{T}_\pi V_\theta - V_\theta\|_{\mu_\pi}^2, \qquad (8)$$

where $\|v\|_{\mu_\pi}^2 = \int v^2(s)\mu_\pi(ds)$, $\mu_\pi$ is the stationary distribution induced by $\pi$, and $\mathbf{\Pi}$ is a projection onto the linear function space w.r.t. $\|\cdot\|_{\mu_\pi}$. We will develop a two time-scale gradient-based approach to minimize the MSPRBE. However, it can be seen that $\max_s V_\theta(s)$ in (7) is not smooth in $\theta$, which is troublesome in both algorithm design and analysis. To solve this issue, we introduce the following smoothed robust Bellman operator $\hat{\mathbf{T}}_\pi$ by smoothing the max with a LSE(LogSumExp):

$$\hat{\mathbf{T}}_\pi V(s) = \mathbb{E}_{A \sim \pi(\cdot|s)} \left[ c(s,A) + \gamma(1-R) \sum_{s' \in \mathcal{S}} p_{s,s'}^A V(s') + \gamma R \cdot \text{LSE}(V) \right], \qquad (9)$$

where $\text{LSE}(V) = \frac{\log\left(\sum_s e^{\varrho V(s)}\right)}{\varrho}$ is the LogSumExp w.r.t. $V$ with a parameter $\varrho > 0$. Note that when $\varrho \to \infty$, the smoothed robust Bellman operator $\hat{\mathbf{T}}_\pi \to \mathbf{T}_\pi$. The LSE operator can also be replaced by some other operator that approximates the max operator and is smooth, e.g., mellow-max [Asadi and Littman, 2017]. In the following, we first show that the fixed point of $\hat{\mathbf{T}}_\pi$ exists for any $\varrho$, and the fixed points converge to the one of $\mathbf{T}_\pi$ for large $\varrho$.

**Theorem 4.** *(1). For any $\varrho$, $\hat{\mathbf{T}}_\pi$ has a fixed point.*

*(2). Let $V_1$ and $V_2$ be the fixed points of $\hat{\mathbf{T}}_\pi$ and $\mathbf{T}_\pi$, respectively. Then*

$$\|V_1 - V_2\|_\infty \le \frac{\gamma R}{1 - \gamma} \frac{\log |\mathcal{S}|}{\varrho} \to 0, \ as \ \varrho \to \infty. \tag{10}$$

We then denote by $J(\theta)$ the smoothed MSPRBE with the LSE operator, and the goal is:

$$\min_\theta J(\theta) = \min_\theta \left\|\mathbf{\Pi}\hat{\mathbf{T}}_\pi V_\theta - V_\theta\right\|_{\mu_\pi}^2. \tag{11}$$

## 5.1   Algorithm Development

In the following, we develop the robust TDC algorithm to solve the problem in (11). We will first derive the gradient of the smoothed MSPRBE, $J(\theta)$, and then design a two time-scale update rule using the weight doubling trick in [Sutton et al., 2009] to solve the double sampling problem. Define $\delta_{s,a,s'}(\theta) \triangleq c(s,a) + \gamma(1 - R)V_\theta(s') + \gamma R\text{LSE}(V_\theta) - V_\theta(s)$, where $\text{LSE}(V_\theta)$ is the LogSumExp function w.r.t. $V_\theta = \theta^\top\phi$. Denote by $C \triangleq \mathbb{E}_{\mu_\pi}\left[\phi_S^\top\phi_S\right]$. Then, $\mathbb{E}_\mu[\delta_{S,A,S'}(\theta)\phi_S] = \Phi^\top D\left(\hat{\mathbf{T}}_\pi V_\theta - V_\theta\right)$, where $D = \text{diag}(\mu_\pi(s_1), \mu_\pi(s_2), ..., \mu_\pi(s_{|\mathcal{S}|}))$ and $\Phi = (\phi_{s_1}, \phi_{s_2}, ..., \phi_{s_{|\mathcal{S}|}})^\top \in \mathbb{R}^{|\mathcal{S}|\times N}$. We know that $\mathbf{\Pi}^\top D\mathbf{\Pi} = D^\top\Phi(\Phi^\top D\Phi)^{-1}\Phi^\top D$ from [Maei, 2011]. Hence we have

$$J(\theta) = \left\|\mathbf{\Pi}\hat{\mathbf{T}}_\pi V_\theta - V_\theta\right\|_{\mu_\pi}^2 = \mathbb{E}_{\mu_\pi}[\delta_{S,A,S'}(\theta)\phi_S]^\top C^{-1}\mathbb{E}_{\mu_\pi}[\delta_{S,A,S'}(\theta)\phi_S]. \tag{12}$$

Then, its gradient can be written as:

$$-\frac{1}{2}\nabla J(\theta) = -\mathbb{E}_{\mu_\pi}[(\nabla\delta_{S,A,S'}(\theta))\phi_S]^\top C^{-1}\mathbb{E}_{\mu_\pi}[\delta_{S,A,S'}(\theta)\phi_S]$$

$$= \mathbb{E}_{\mu_\pi}[\delta_{S,A,S'}(\theta)\phi_S] - \gamma\mathbb{E}_{\mu_\pi}\left[\left((1 - R)\phi_{S'} + R \cdot \nabla\text{LSE}(V_\theta)\right)\phi_S^\top\right]\omega(\theta),$$

where $\omega(\theta) = C^{-1}\mathbb{E}_{\mu_\pi}[\delta_{S,A,S'}(\theta)\phi_S]$. It can be seen that to obtain an unbiased estimate of $\nabla J(\theta)$, two independent samples are needed as there exists a multiplication of two expectations, which is not applicable when there is only one sample trajectory. We then utilize the weight doubling trick in [Sutton et al., 2009], and design the robust TDC algorithm in Algorithm 2. Specifically, we introduce a fast time scale to estimate $\omega(\theta)$, and a slow time scale to estimate $\nabla J(\theta)$. Denote the projection by $\mathbf{\Pi}_K(x) \triangleq \arg\min_{\|y\|\le K}\|y - x\|$ for any $x \in \mathbb{R}^N$. Our robust TDC algorithm in Algorithm 2 can be implemented in an online and incremental fashion. If the uncertainty set becomes a singleton, i.e., $R = 0$, then Algorithm 2 reduces to the vanilla TDC algorithm.

## 5.2   Finite-Time Error Bound of Robust TDC

Unlike the vanilla TDC algorithm, $J(\theta)$ here is non-convex. Therefore, we are interested in the convergence to stationary points, i.e., the rate of $\|\nabla J(\theta)\| \to 0$. We first make some standard assumptions which are commonly used in RL algorithm analysis, e.g., [Wang and Zou, 2020, Kaledin et al., 2020, Xu et al., 2019, Srikant and Ying, 2019, Bhandari et al., 2018].

**Assumption 2** (Bounded feature). $\|\phi_s\|_2 \le 1, \forall s \in \mathcal{S}$.

**Assumption 3** (Bounded cost function). $|c(s,a)| \le c_{\max}, \forall s \in \mathcal{S}$ and $a \in \mathcal{A}$.

**Assumption 4** (Problem solvability). *The matrix $C = \mathbb{E}_{\mu_\pi}[\phi_S\phi_S^\top]$ is non-singular with $\lambda > 0$ being its smallest eigenvalue.*

---

**Algorithm 2** Robust TDC with Linear Function Approximation

---

**Input**: $T, \alpha, \beta, \varrho, \phi_i$ for $i = 1, ..., N$, projection radius $K$
**Initialization**: $\theta_0, w_0, s_0$
 1: Choose $W \sim \text{Uniform}(0, 1, ..., T - 1)$
 2: **for** $t = 0, 1, 2, ..., W - 1$ **do**
 3:    Take action according to $\pi(\cdot|s_t)$ and observe $s_{t+1}$ and $c_t$
 4:    $\phi_t \leftarrow \phi_{s_t}$
 5:    $\delta_t(\theta_t) \leftarrow c_t + \gamma(1 - R)V_{\theta_t}(s_{t+1}) + \gamma R \frac{\log(\sum_s e^{e^{\theta^\top \phi_s}})}{\varrho} - V_{\theta_t}(s_t)$
 6:    $\theta_{t+1} \leftarrow \mathbf{\Pi}_K \left( \theta_t + \alpha \left( \delta_t(\theta_t)\phi_t - \gamma \left( (1 - R)\phi_{t+1} + R \sum_{s \in \mathcal{S}} \left( \frac{e^{\varrho V_\theta(s)}\phi_s}{\sum_{j \in \mathcal{S}} e^{\varrho V_\theta(j)}} \right) \right) \phi_t^\top \omega_t \right) \right)$
 7:    $\omega_{t+1} \leftarrow \mathbf{\Pi}_K(\omega_t + \beta(\delta_t(\theta_t) - \phi_t^\top \omega_t)\phi_t)$
 8: **end for**
**Output**: $\theta_W$

---

**Assumption 5** (Geometric uniform ergodicity). *There exist some constants $m > 0$ and $\rho \in (0, 1)$ such that for any $t > 0$, $\max_{s \in \mathcal{S}} d_{TV}(\mathbb{P}(s_t|s_0 = s), \mu_\pi) \leq m\rho^t$.*

In the following theorem, we characterize the finite-time error bound for the convergence of our robust TDC algorithm. Here we only provide the order of the bounds in terms of $T$. The explicit bounds can be found in (129) in Appendix D.3.

**Theorem 5.** *Consider the following step-sizes: $\beta = \mathcal{O}\left(\frac{1}{T^b}\right)$, and $\alpha = \mathcal{O}\left(\frac{1}{T^a}\right)$, where $\frac{1}{2} < a \leq 1$ and $0 < b \leq a$. Then we have that*

$$\mathbb{E}[\|\nabla J(\theta_W)\|^2] = \mathcal{O}\left( \frac{1}{T\alpha} + \alpha \log(1/\alpha) + \frac{1}{T\beta} + \beta \log(1/\beta) \right), \qquad (13)$$

*If we further let $a = b = 0.5$, then $\mathbb{E}[\|\nabla J(\theta_W)\|^2] = \mathcal{O}\left(\frac{\log T}{\sqrt{T}}\right)$.*

The robust TDC has a matching complexity with the vanilla TDC with non-linear function approximation [Wang et al., 2021], but provides the additional robustness to model uncertainty. It does not need to relax the uncertainty set like in [Roy et al., 2017], and our convergence results do not need a condition on the discount factor.

## 6 Experiments

### 6.1 Robust Q-Learning

In this section, we compare our robust Q-learning with the vanilla non-robust Q-learning. We use OpenAI gym framework [Brockman et al., 2016], and consider two different problems: Frozen lake and Cart-Pole. One more example of the taxi problem is given in the appendix. To demonstrate the robustness, the policy is learned in a perturbed MDP, and is then tested on the true unperturbed MDP. Specifically, during the training, we set a probability $p$ such that after the agent takes an action, with probability $p$, the state transition is uniformly over $\mathcal{S}$, and with probability $1 - p$ the state transition is according to the true unperturbed transition kernel. The behavior policy for all the experiments below is set to be a uniform distribution over the action space given any state, i.e., $\pi_b(a|s) = \frac{1}{|\mathcal{A}|}$ for any $s \in \mathcal{S}$ and $a \in \mathcal{A}$. We then evaluate the performance of the obtained policy in the unperturbed environment. At each time t, the policy we evaluate is the greedy-policy w.r.t. the current estimate of the Q-function, i.e., $\pi_t(s) = \arg\max_a Q_t(s, a)$. A Monte-Carlo method with horizon 100 is used to evaluate the accumulated discounted reward of the learned policy on the unperturbed MDP. We take the average over 30 trajectories. More details are provided in the appendix.

In Figure 1 and Figure 2, we plot the accumulated discounted reward of both algorithms under different $p$ and $R$ for both problems. The upper and lower envelopes of the curves correspond to the 95 and 5 percentiles of the 30 trajectories, respectively. It can be seen that overall our robust Q-learning algorithm achieves a much higher reward than the vanilla Q-learning. This demonstrates the robustness of our robust Q-learning algorithm to model uncertainty. Moreover, as $p$ and $R$ getting

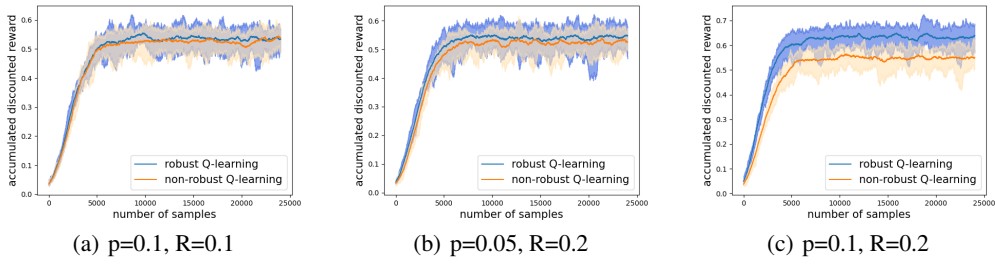

Figure 1: **FrozenLake-v0**: robust Q-learning v.s. non-robust Q-learning.

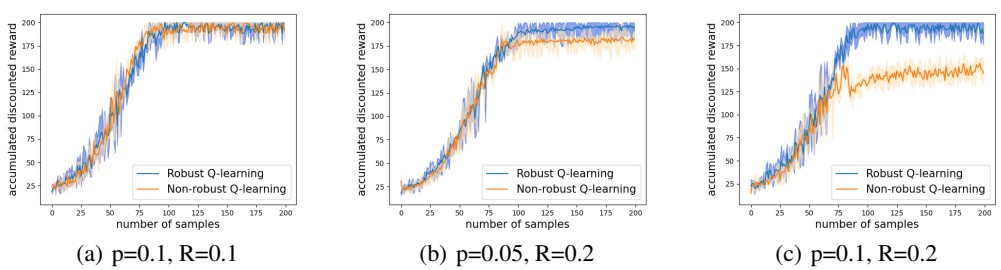

Figure 2: **CartPole-v0**: robust Q-learning v.s. non-robust Q-learning.

larger, i.e., as the MDP that we learn the policy deviates from the MDP we test the policy, the advantage of our robust Q-learning algorithm is getting more significant compared to the vanilla Q-learning algorithm.

## 6.2 Robust TDC with Linear Function Approximation

In this section we compare our robust TDC with the vanilla non-robust TDC with linear function approximation on the $4 \times 4$ Frozen Lake problem. The problem setting is the same as the one in Section 6.1. More details about the experiment setup are provided in the appendix.

We implement the two algorithms using samples from the perturbed MDP both for 30 times, and obtain 30 sequences of $\{\theta_t^i\}_{t=1}^{\infty}$, $i = 1, 2, ..., 30$. We then compute the squared gradient norm $\|\nabla J(\theta)\|^2$ on the true unperturbed MDP, and see whether $\{\theta_t^i\}_{t=1}^{\infty}$ converges to some stationary points on the true unperturbed MDP. In Fig. 3, we plot the average squared gradient norm $\|\nabla J(\theta)\|^2$ for different $p$ and $R$. The upper and lower envelops are the 95 and 5 percentiles of the 30 curves. It can be seen that our robust TDC converges much faster than vanilla TDC, and as the model mismatch between the training and test MDPs enlarges, the vanilla TDC may diverge (Fig. 3(c)), while our robust TDC still converges to some stationary point. Also, the robust TDC has a much smaller variance, which indicates a much stable behavior under model uncertainty.

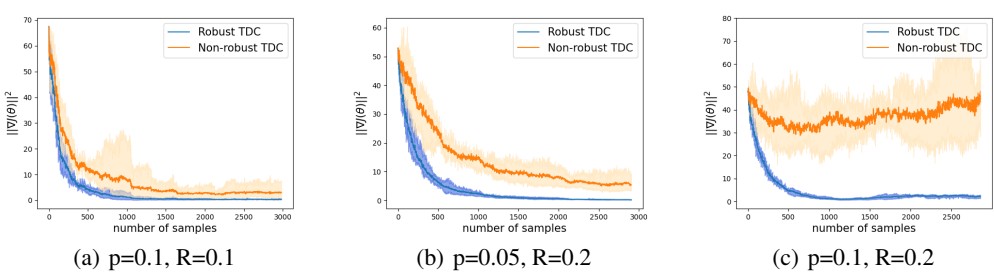

Figure 3: **FrozenLake-v0**: $\|\nabla J(\theta)\|^2$ of robust TDC and non-robust TDC.

### 6.3 Comparison with The Adversarial Training Approach

We also compare our robust Q-learning with Robust Adversarial Reinforcement Learning (RARL) in [Pinto et al., 2017]. To apply their algorithm to our problem setting, we model the nature as an adversarial player, and its goal is to minimize the reward that the agent receives. The action space $\mathcal{A}_{ad}$ of the nature is set to be the state space $\mathcal{A}_{ad} \triangleq \mathcal{S}$. Then the perturbed training environment can be viewed as an adversarial model: both the agent and the adversary take actions $a_a, a_{ad}$, then the environment will transit to state $a_{ad}$ with probability $R$ and transit following the unperturbed MDP $p_s^{a_a}$ with probability $1 - R$. The goal of the maximize its accumulated reward, while the goal of the natural is to minimize it.

Following the RARL algorithm [Pinto et al., 2017], in each iteration of the training, we first fix the adversarial policy and use Q-learning to optimize the agent's policy and obtain the Q-table $Q_t$. Then we fix the agent's policy and optimize the adversarial policy.

After each training iteration, we test the performance of the greedy policies w.r.t. Q-tables obtained from robust Q-learning and RARL. The testing environment is set to be the worst-case, i.e., after the agent takes an action, the environment transits to the state which has the minimal value function ($\arg\min_{s \in \mathcal{S}} V_t(s)$) with probability $p$. We plot the accumulated discounted rewards of both algorithms against number of training iterations under different parameters. We set $\alpha = 0.2$ and $\gamma = 0.9$. It can be seen from Fig. 4 that our robust Q-learning achieves a higher accumulative reward, and thus is more robust that the RARL algorithm in [Pinto et al., 2017]. Also our robust Q-learning is more stable during training, i.e., the variance is smaller.

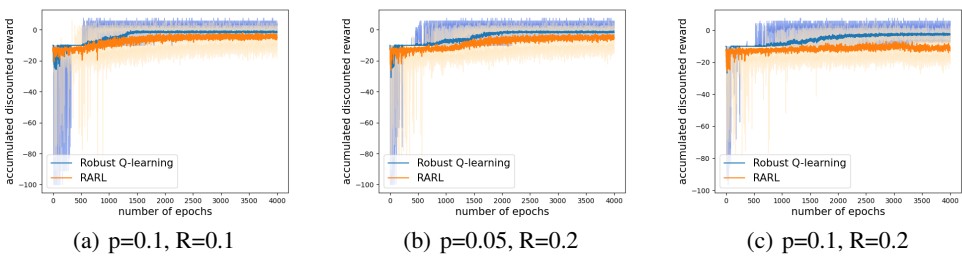

(a) p=0.1, R=0.1      (b) p=0.05, R=0.2      (c) p=0.1, R=0.2

Figure 4: **Taxi-v3**: robust Q-learning v.s. RARL.

## 7 Conclusion

In this paper, we develop a novel approach for solving model-free robust RL problems with model uncertainty. Our algorithms can be implemented in an online and incremental fashion, do not require additional memory than their non-robust counterparts. We theoretically proved the convergence of our algorithms under no additional assumption on the discount factor, and further characterized their finite-time error bounds, which match with their non-robust counterparts (within a constant factor). Our approach can be readily extended to robustify TD, SARSA and other GTD algorithms. **Limitations:** It is also of future interest to investigate robustness to reward uncertainty, and other types of uncertainty sets, e.g., ones defined by KL divergence, Wasserstein distance and total variation. **Negative societal impact:** To the best of the authors' knowledge, this study does not have any potential negative impact on the society.

## 8 Acknowledgment

The work of Y. Wang and S. Zou was supported by the National Science Foundation under Grants CCF-2106560 and CCF- 2007783.

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
