# Supplementary Materials

## A    Proof of Theorem 2: Asymptotic Convergence of Robust Q-Learning

In this section we show that the robust Q-learning converges exactly to the optimal robust Q function $Q^*$. Recall that the optimal robust Q function $Q^*$ is the solution to the robust Bellman operator $\mathbf{T}$:

$$Q^*(s,a) = c(s,a) + \gamma\sigma_{\mathcal{P}_s^a}((\min_{a\in\mathcal{A}} Q^*(s_1,a), \min_{a\in\mathcal{A}} Q^*(s_2,a), ..., \min_{a\in\mathcal{A}} Q^*(s_{|\mathcal{S}|},a))^\top). \quad (14)$$

It can be shown that the estimated update is an unbiased estimation of $\mathbf{T}$. More specifically,

$$\begin{aligned}
\mathbf{T}Q(s,a) &= c(s,a) + \gamma\sigma_{\mathcal{P}_s^a}(V) \\
&= c(s,a) + \gamma(1-R)(p_s^a)^\top V + R\max_{s'} V(s') \\
&= c(s,a) + \gamma(1-R)\sum_{s'}(p_{s,s'}^a)V(s') + R\max_{s'} V(s') \\
&= c(s,a) + \gamma\sum_{s'} p_{s,s'}^a \left((1-R)(\mathbb{1}_{s'})^\top V + R\max_q q^\top V\right),
\end{aligned} \quad (15)$$

which is the expectation of the estimated update in line 5 of Algorithm 1.

### A.1    Robust Bellman operator is a contraction

It was shown in [Iyengar, 2005, Roy et al., 2017] that the robust Bellman operator is a contraction. Here, for completeness, we include the proof for our R-contamination uncertainty set. More specifically,

$$\begin{aligned}
&|\mathbf{T}Q(s,a) - \mathbf{T}Q'(s,a)| \\
&= |c(s,a) + \gamma\sigma_{\mathcal{P}_s^a}(V) - c(s,a) - \gamma\sigma_{\mathcal{P}_s^a}(V')| \\
&= \gamma|\sigma_{\mathcal{P}_s^a}(V) - \sigma_{\mathcal{P}_s^a}(V')| \\
&= \gamma|\max_q\{(1-R)(p_s^a)^\top V + Rq^\top V\} - \max_{q'}\{(1-R)(p_s^a)^\top V' + Rq'^\top V'\}| \\
&= \gamma\left|\sum_{s'\in\mathcal{S}} p_{s,s'}^a((1-R)V(s')) + R\max_{s'} V(s') - \sum_{s'\in\mathcal{S}} p_{s,s'}^a((1-R)V'(s')) - R\max_{s'} V'(s')\right| \\
&= \gamma\left|\sum_{s'\in\mathcal{S}} p_{s,s'}^a(1-R)(V(s') - V'(s')) + R(\max_{s'} V(s') - \max_{s'} V'(s'))\right| \\
&\leq \gamma\left|\sum_{s'\in\mathcal{S}} p_{s,s'}^a(1-R)\left(\min_a Q(s',a) - \min_b Q'(s',b)\right)\right| + \gamma R(|\max_{s'} V(s') - \max_{s'} V'(s')|) \\
&\leq \gamma\sum_{s'\in\mathcal{S}} p_{s,s'}^a(1-R)\left|\left(\min_a Q(s',a) - \min_b Q'(s',b)\right)\right| + \gamma R\max_s|(V(s) - V'(s))| \\
&\overset{(a)}{\leq} \gamma\sum_{s'\in\mathcal{S}} p_{s,s'}^a(1-R)\|Q - Q'\|_\infty + \gamma R\|Q - Q'\|_\infty \\
&\leq \gamma\|Q - Q'\|_\infty,
\end{aligned} \quad (16)$$

where $(a)$ can be shown as below. Assume that $a_1 = \arg\min_a Q(s',a)$ and $b_1 = \arg\min_a Q'(s',a)$. Then if $Q(s',a_1) > Q'(s',b_1)$, then

$$|Q(s',a_1) - Q'(s',b_1)| = Q(s',a_1) - Q'(s',b_1) \leq Q(s',b_1) - Q'(s',b_1) \leq \|Q - Q'\|_\infty. \quad (17)$$

Similarly, it can also be shown when $Q(s',a_1) \leq Q'(s',b_1)$, and hence the inequality $(a)$ holds.

### A.2 Asymptotic Convergence of Robust Q-Leaning

With the definition of $\mathbf{T}$, the update (5) of robust Q-learning can be re-written as a stochastic approximation:

$$Q_{t+1}(s_t, a_t) = (1 - \alpha_t)Q_t(s_t, a_t) + \alpha_t(\mathbf{T}Q_t(s_t, a_t) + \eta_t(s_t, a_t, s_{t+1})), \tag{18}$$

where the noise term is

$$\eta_t(s_t, a_t, s_{t+1}) = c(s_t, a_t) + \gamma R \max_s V_t(s) + \gamma(1 - R)V_t(s_{t+1}) - \mathbf{T}Q_t(s_t, a_t). \tag{19}$$

From (15), we have that

$$\mathbb{E}[\eta_t(S_t, A_t, S_{t+1})|S_t = s_t, A_t = a_t] = 0. \tag{20}$$

The variance can be bounded by

$$\mathbb{E}[(\eta_t(S_t, A_t, S_{t+1}))^2] \leq \gamma^2 (1 - R)^2 (\max_{s,a} Q_t^2(s, a)), \tag{21}$$

where the last inequality is from $V_t(s_{t+1}) \leq \max_s V_t(s) \leq \max_{s,a} Q_t(s, a)$. Thus the noise term $\eta_t$ has zero mean and bounded variance. From [Borkar and Meyn, 2000], we know that the stochastic approximation (18) converges to the fixed point of $\mathbf{T}$, i.e., $Q^*$. Hence we showed that robust Q-learning converges to optimal optimal robust Q function $Q^*$ with probability 1.

## B  Finite-Time Analysis of Robust Q-Learning

In this section, we develop the finite-time analysis of the Algorithm 1.

### B.1  Notations

We first introduce some notations. For a vector $v = (v_1, v_2, ..., v_n)$, we denote the entry wise absolute value $(|v_1|, ..., |v_n|)$ by $|v|$. For a sample $O_t = (s_t, a_t, s_{t+1})$, define $\Lambda_{t+1} \in \mathbb{R}^{|\mathcal{S}||\mathcal{A}| \times |\mathcal{S}||\mathcal{A}|}$ as

$$\Lambda_{t+1}((s,a),(s',a')) = \begin{cases} \alpha, & \text{if } (s,a) = (s',a') = (s_t, a_t), \\ 0, & \text{otherwise.} \end{cases} \tag{22}$$

Also we define the sample transition matrix $P_{t+1} \in \mathbb{R}^{|\mathcal{S}||\mathcal{A}| \times |\mathcal{S}|}$ as

$$P_{t+1}((s,a),s') = \begin{cases} 1, & \text{if } (s,a,s') = O_t, \\ 0, & \text{otherwise.} \end{cases} \tag{23}$$

We also define the transition kernel matrix $P \in \mathbb{R}^{|\mathcal{S}||\mathcal{A}| \times |\mathcal{S}|}$ as

$$P((s,a),s') = p_{s,s'}^a. \tag{24}$$

We use $Q_t \in \mathbb{R}^{|\mathcal{S}||\mathcal{A}|}$ and $V_t \in \mathbb{R}^{|\mathcal{S}|}$ to denote the vectors of value functions. Denote the cost function $c \in \mathbb{R}^{|\mathcal{S}||\mathcal{A}|}$ with entry $c(s,a)$ being the cost received at $(s,a)$. Then the update of robust Q-learning (5) can be written in matrix form as

$$Q_t = (I - \Lambda_t)Q_{t-1} + \Lambda_t \Big(c + \gamma(1 - R)P_t V_{t-1} + \gamma R \max_{s \in \mathcal{S}} V_{t-1}(s)P_t \mathbf{1}\Big), \tag{25}$$

where $\mathbf{1}$ denotes the vector $(1, 1, 1, ..., 1)^\top \in \mathbb{R}^{|\mathcal{S}|}$. The robust Bellman equation can be written as

$$Q^* = c + \gamma(1 - R)PV^* + \gamma R \max_{s \in \mathcal{S}} V^*(s)P\mathbf{1}. \tag{26}$$

### B.2  Analysis

Define $\psi_t = Q_t - Q^*$, then by (25) and (26), we have that

$$\begin{aligned} \psi_t &= Q_t - Q^* \\ &= (I - \Lambda_t)Q_{t-1} + \Lambda_t(c + \gamma(1 - R)P_t V_{t-1} + \gamma R \max_{s \in \mathcal{S}} V_{t-1}(s)P_t \mathbf{1}) - Q^* \end{aligned}$$

$$= (I - \Lambda_t)(Q_{-1} - Q^*) + \Lambda_t(c + \gamma(1-R)P_t V_{t-1} + \gamma R \max_{s \in \mathcal{S}} V_{t-1}(s)P_t \mathbf{1} - Q^*)$$

$$= (I - \Lambda_t)\psi_{t-1} + \Lambda_t(\gamma(1-R)P_t V_{t-1} + \gamma R \max_{s \in \mathcal{S}} V_{t-1}(s)P_t \mathbf{1} - \gamma(1-R)PV^*$$

$$- \gamma R \max_{s \in \mathcal{S}} V^*(s)P\mathbf{1})$$

$$= (I - \Lambda_t)\psi_{t-1} + \gamma(1-R)\Lambda_t \underbrace{(P_t V_{t-1} - PV^*)}_{k_1}$$

$$+ \gamma R \Lambda_t \underbrace{(\max_{s \in \mathcal{S}} V_{t-1}(s)P_t \mathbf{1} - \max_{s \in \mathcal{S}} V^*(s)P\mathbf{1})}_{k_2}. \tag{27}$$

The term $k_1$ can be written as

$$P_t V_{t-1} - PV^* = P_t V_{t-1} - P_t V^* + P_t V^* - PV^* = P_t(V_{t-1} - V^*) + (P_t - P)V^*. \tag{28}$$

Similarly, we have that

$$k_2 = \left(\max_{s \in \mathcal{S}} V_{t-1}(s) - \max_{s \in \mathcal{S}} V^*(s)\right)P_t \mathbf{1} + \max_{s \in \mathcal{S}} V^*(s)(P_t - P)\mathbf{1}. \tag{29}$$

Hence (27) can be written as

$$\psi_t = Q_t - Q^*$$

$$= (I - \Lambda_t)\psi_{t-1} + \gamma(1-R)\Lambda_t(P_t(V_{t-1} - V^*) + (P_t - P)V^*)$$

$$+ \gamma R \Lambda_t \left( \left(\max_{s \in \mathcal{S}} V_{t-1}(s) - \max_{s \in \mathcal{S}} V^*(s)\right)P_t \mathbf{1} + \max_{s \in \mathcal{S}} V^*(s)(P_t - P)\mathbf{1} \right)$$

$$= (I - \Lambda_t)\psi_{t-1} + \left( \gamma(1-R)\Lambda_t(P_t - P)V^* + \gamma R \Lambda_t(\max_{s \in \mathcal{S}} V^*(s)(P_t - P)\mathbf{1}) \right)$$

$$+ \left( \gamma(1-R)\Lambda_t(P_t(V_{t-1} - V^*)) + \gamma R \Lambda_t \left( \left(\max_{s \in \mathcal{S}} V_{t-1}(s) - \max_{s \in \mathcal{S}} V^*(s)\right)P_t \mathbf{1} \right) \right). \tag{30}$$

By applying (30) recursively, we have that

$$\psi_t = \underbrace{\prod_{j=1}^{t}(I - \Lambda_j)\psi_0}_{k_{1,t}}$$

$$+ \underbrace{\gamma(1-R)\sum_{i=1}^{t}\prod_{j=i+1}^{t}(I - \Lambda_j)\Lambda_i(P_i - P)V^* + \gamma R \sum_{i=1}^{t}\prod_{j=i+1}^{t}(I - \Lambda_j)\Lambda_i \max_{s \in \mathcal{S}} V^*(s)(P_i - P)\mathbf{1}}_{k_{2,t}}$$

$$+ \underbrace{\gamma(1-R)\sum_{i=1}^{t}\prod_{j=i+1}^{t}(I - \Lambda_j)\Lambda_i P_i(V_{i-1} - V^*) + \gamma R \sum_{i=1}^{t}\prod_{j=i+1}^{t}(I - \Lambda_j)\Lambda_i(\max_{s \in \mathcal{S}} V_{i-1}(s) - \max_{s \in \mathcal{S}} V^*(s))P_i \mathbf{1}}_{k_{3,t}}.$$

$$\tag{31}$$

We then bound terms $k_{i,t}$ separately.

**Lemma 1.** *Define $t_{frame} = \frac{443 t_{mix}}{\mu_{\min}} \log \frac{4|\mathcal{S}||\mathcal{A}|T}{\delta}$. Then with probability at least $1 - \delta$, for any $(s,a) \in \mathcal{S} \times \mathcal{A}$ and any $t \geq t_{frame}$, $k_{1,t}$ can be bounded as*

$$|k_{1,t}| \leq (1 - \alpha)^{\frac{t\mu_{\min}}{2}} \|\psi_0\|_\infty \mathbf{1}; \tag{32}$$

*and for $t < t_{frame}$,*

$$|k_{1,t}| \leq \|\psi_0\|_\infty \mathbf{1}. \tag{33}$$

*Proof.* First note that the $(s,a)$-entry of $k_{1,t}$ can be written as

$$k_{1,t}(s,a) = (1 - \alpha)^{K_t(s,a)}\psi_0(s,a), \tag{34}$$

where $K_t(s,a)$ denotes the times that the sample trajectory visits $(s,a)$ before the time step $t$. We introduce a lemma from [Li et al., 2020] first:

**Lemma 2.** *(Lemma 5 [Li et al., 2020]) For a time-homogeneous and uniformly ergodic Markov chain with state space $\mathcal{X}$ and any $0 < \delta < 1$, if $t \geq \frac{443 t_{mix}}{\mu_{\min}} \log \frac{|\mathcal{X}|}{\delta}$, then for any $y \in \mathcal{X}$,*

$$\mathbb{P}_{X_1=y}\left\{\exists x \in \mathcal{X} : \sum_{j=1}^{t} \mathbb{1}X_j = x \leq \frac{t\mu(x)}{2}\right\} \leq \delta, \tag{35}$$

*where $t_{mix} = \min\left\{t : \max_{x \in \mathcal{X}} d_{TV}(\mu, P^t(\cdot|x)) \leq \frac{1}{4}\right\}$; $\mu$ is the stationary distribution of the Markov chain, and $\mu_{\min} \triangleq \min_{x \in \mathcal{X}} \mu(x)$.*

From this lemma, we know that for any $(s,a) \in \mathcal{S} \times \mathcal{A}$ and any $t \geq \frac{443 t_{mix}}{\mu_{\min}} \log \frac{4|\mathcal{S}||\mathcal{A}|T}{\delta}$, we have that

$$K_t(s,a) \geq \frac{t\mu_{\min}}{2}, \tag{36}$$

with probability at least $1 - \delta$.

Thus (34) can be bounded as

$$|k_{1,t}(s,a)| \leq (1-\alpha)^{\frac{t\mu_{\min}}{2}}|\psi_0(s,a)| \tag{37}$$

with probability at least $1 - \delta$ for any $(s,a) \in \mathcal{S} \times \mathcal{A}$ and any $t \geq \frac{443 t_{mix}}{\mu_{\min}} \log \frac{4|\mathcal{S}||\mathcal{A}|T}{\delta}$, which shows the claim.

For $t < t_{\text{frame}}$, the bound is obvious by noting that $\|I - \Lambda_j\| \leq 1$. $\qquad\square$

**Lemma 3.** *There exists some constant $\hat{c}$, such that for any $\delta < 1$ and any $t \leq T$ that satisfies $0 < \alpha \log \frac{|\mathcal{S}||\mathcal{A}|T}{\delta} < 1$, with probability at least $1 - \frac{\delta}{|\mathcal{S}||\mathcal{A}|T}$,*

$$|k_{2,t}| \leq 5\gamma\hat{c}\sqrt{\alpha \log \frac{T|\mathcal{S}||\mathcal{A}|}{\delta}}\|V^*(s)\|_\infty \mathbf{1}, \tag{38}$$

*Proof.* Recall that

$$k_{2,t} = \gamma(1-R)\sum_{i=1}^{t}\prod_{j=i+1}^{t}(I-\Lambda_j)\Lambda_i(P_i-P)V^* + \gamma R\sum_{i=1}^{t}\prod_{j=i+1}^{t}(I-\Lambda_j)\Lambda_i(P_i-P)w^*, \tag{39}$$

where $w^* \triangleq \max_{s \in \mathcal{S}} V^*(s)\mathbf{1}$. Then the $(s,a)$-th entry of $k_{2,t}$ can be written as

$$k_{2,t}(s,a) = \gamma(1-R)\sum_{i=1}^{K_t(s,a)}\alpha(1-\alpha)^{K_t(s,a)-i}(P_{t_i+1}(s,a) - P(s,a))V^*$$

$$+ \gamma R\sum_{i=1}^{K_t(s,a)}\alpha(1-\alpha)^{K_t(s,a)-i}(P_{t_i+1}(s,a) - P(s,a))w^*, \tag{40}$$

where $t_i(s,a)$ is the time step when the trajectory visits $(s,a)$ for the $i$-th time. We define $\text{Var}_P(V) \in \mathbb{R}^{|\mathcal{S}||\mathcal{A}|}$ being a vector, where $\text{Var}_P(V)(s,a) = \sum_{s' \in \mathcal{S}} p_{s,s'}^a(V(s')^2) - (\sum_{s' \in \mathcal{S}} p_{s,s'}^a V(s'))^2 \triangleq \text{Var}_{P_s^a}[V]$ for any $V \in \mathbb{R}^{|\mathcal{S}|}$.

From Section E.1 in [Li et al., 2020], we know that

$$\text{Var}\left[\sum_{i=1}^{K}\alpha(1-\alpha)^{K-i}(P_{t_i+1}(s,a) - P(s,a))V^*\right] = \alpha\text{Var}_{P_s^a}[V^*] \triangleq \sigma_K^2 \tag{41}$$

for some constant $\sigma_K^2$ and any $K \leq T$. Moreover, note that

$$\text{Var}\left[\sum_{i=1}^{K}\alpha(1-\alpha)^{K-i}(P_{t_i+1}(s,a) - P(s,a))w^*\right]$$

$$\overset{(a)}{=} \sum_{i=1}^{K} \alpha^2 (1-\alpha)^{2K-2i} \text{Var}[(P_{t_i+1}(s,a) - P(s,a))w^*]$$

$$\overset{(b)}{=} \sum_{i=1}^{K} \alpha^2 (1-\alpha)^{2K-2i} \text{Var}[\max_s V^*(s)((P_{t_i+1}(s,a) - P(s,a))\mathbf{1})]$$

$$= 0, \tag{42}$$

where equation $(a)$ is due to the fact that $\{P_{t_1+1}(s,a), P_{t_2+1}(s,a), ..., P_{t_i+1}(s,a)\}_{i \in \mathbb{N}}$ are independent (Equation $(101)$ in [Li et al., 2020]), $(b)$ is from the definition of $\omega^*$, and the last equation is because the sum of each entries of $P_{t_i+1}(s,a) - P(s,a)$ is 0.

the last equality is due to the fact that every entries of $w^*$ are the same and hence $\text{Var}_{P_s^a}[w^*] = 0$.

Additionally, we have that

$$\left\| \alpha(1-\alpha)^{K-i}(P_{t_i+1}(s,a) - P(s,a))V^* \right\|_\infty \le 2\alpha\|V^*(s)\|_\infty \triangleq D, \tag{43}$$

where we denote the bound by $D$. Also,

$$\left\| \alpha(1-\alpha)^{K-i}(P_{t_i+1}(s,a) - P(s,a))w^* \right\|_\infty \le D. \tag{44}$$

Hence from the Bernstein inequality ([Li et al., 2020]), we have that

$$|k_{2,t}(s,a)|$$

$$\le \gamma(1-R)\hat{c}\left( \sqrt{\sigma_K^2 \log\left(\frac{T|\mathcal{S}||\mathcal{A}|}{\delta}\right)} + D\log\frac{T|\mathcal{S}||\mathcal{A}|}{\delta} \right) + \gamma R\hat{c}\left( D\log\frac{T|\mathcal{S}||\mathcal{A}|}{\delta} \right)$$

$$\le 5\gamma\hat{c}\sqrt{\alpha \log\frac{T|\mathcal{S}||\mathcal{A}|}{\delta}} \|V^*(s)\|_\infty, \tag{45}$$

for some constant $\hat{c}$ with probability at least $1 - \frac{\delta}{|\mathcal{S}||\mathcal{A}|T}$, and the last step is due to the fact that $\text{Var}_{P_s^a}[V^*] \le \|V^*\|_\infty^2$ and $\alpha \log\frac{|\mathcal{S}||\mathcal{A}|T}{\delta} < 1$. This hence completes the proof. $\qquad \square$

**Lemma 4.** *For any* $t \ge T$,

$$|k_{3,t}| \le \gamma \sum_{i=1}^{t} \|\psi_{i-1}\|_\infty \prod_{j=i+1}^{t} (I - \Lambda_j)(\Lambda_i)\mathbf{1}. \tag{46}$$

*Proof.* First note that for any $i$,

$$\|P_i(V_{i-1} - V^*)\|_\infty \le \|P_i\|_1 \|V_{i-1} - V^*\|_\infty = \|V_{i-1} - V^*\|_\infty \le \|\psi_{i-1}\|_\infty, \tag{47}$$

where the last inequality is from

$$\|V_{i-1} - V^*\|_\infty = \max_s |V_{i-1}(s) - V^*(s)| = |V_{i-1}(s^*) - V^*(s^*)|$$

$$= |\min_a Q_{i-1}(s^*,a) - \min_b Q^*(s^*,b)| \le \|Q_{i-1} - Q^*\|_\infty, \tag{48}$$

where $s^* = \arg\max |V_{i-1}(s) - V^*(s)|$. Similarly,

$$\left\| (\max_{s \in \mathcal{S}} V_{i-1}(s) - \max_{s \in \mathcal{S}} V^*(s))P_i \mathbf{1} \right\|_\infty \le |\max_{s \in \mathcal{S}} V_{i-1}(s) - \max_{s \in \mathcal{S}} V^*(s)| \le \|\psi_{i-1}\|_\infty, \tag{49}$$

where the last inequality is from $|\max_{s \in \mathcal{S}} V_{i-1}(s) - \max_{s \in \mathcal{S}} V^*(s)| \le \|V_{i-1} - V^*\|_\infty \le \|Q_{i-1} - Q^*\|_\infty$. Hence $K_{3,t}$ can be bounded as

$$|k_{3,t}| \le \gamma \sum_{i=1}^{t} \|\psi_{i-1}\|_\infty \prod_{j=i+1}^{t} (I - \Lambda_j)(\Lambda_i)\mathbf{1}. \tag{50}$$

$\qquad \square$

Now combine the bounds for terms $k_{1,t}$, $k_{2,t}$ and $k_{3,t}$, we have the bound on $\psi_t$ as follows.

For $t < t_{\text{frame}}$, we have that

$$\|\psi_t\|_\infty \le \|\psi_0\|_\infty \mathbf{1} + 5\gamma\hat{c}\sqrt{\alpha \log \frac{T|\mathcal{S}||\mathcal{A}|}{\delta}}\|V^*(s)\|_\infty \mathbf{1}$$
$$+ \gamma \sum_{i=1}^{t} \|\psi_{i-1}\|_\infty \prod_{j=i+1}^{t} (I - \Lambda_j)(\Lambda_i)\mathbf{1}; \tag{51}$$

and for $t \ge t_{\text{frame}}$, we have that

$$\|\psi_t\|_\infty \le (1-\alpha)^{\frac{t\mu_{\min}}{2}}\|\psi_0\|_\infty \mathbf{1} + 5\gamma\hat{c}\sqrt{\alpha \log \frac{T|\mathcal{S}||\mathcal{A}|}{\delta}}\|V^*(s)\|_\infty \mathbf{1}$$
$$+ \gamma \sum_{i=1}^{t} \|\psi_{i-1}\|_\infty \prod_{j=i+1}^{t} (I - \Lambda_j)(\Lambda_i)\mathbf{1}. \tag{52}$$

This bound exactly matches the bound in Equation (42) in [Li et al., 2020] and hence the remaining proof for Theorem 3 can be obtained by following the proof in [Li et al., 2020]. We omit the remaining proof and only state the result.

**Theorem 6.** *Define*

$$t_{th} = \max\left\{\frac{2\log\frac{1}{(1-\gamma)^2\epsilon}}{\alpha\mu_{\min}}, t_{frame}\right\}; \tag{53}$$

$$\mu_{frame} = \frac{1}{2}\mu_{\min}t_{frame}; \tag{54}$$

$$\rho = (1-\gamma)(1-(1-\alpha)^{\mu_{frame}}), \tag{55}$$

*then for any $\delta < 1$ and any $\epsilon < \frac{1}{1-\gamma}$, there exists a universal constant $\hat{c}$ and $c_0$ (determined by $\hat{c}$), such that with probability at least $1 - 6\delta$, the following bound holds for any $t < T$:*

$$\|Q_t - Q^*\|_\infty \le \frac{(1-\rho)^k\|Q_0 - Q^*\|_\infty}{1-\gamma} + \frac{5\hat{c}\gamma}{1-\gamma}\sqrt{\alpha\log\frac{|\mathcal{S}||\mathcal{A}|T}{\delta}} + \epsilon, \tag{56}$$

*where $k = \max\left\{0, \lfloor\frac{t-t_{th}}{t_{frame}}\rfloor\right\}$, as long as*

$$T \ge c_0\left(\frac{1}{\mu_{min}(1-\gamma)^5\epsilon^2} + \frac{t_{mix}}{\mu_{min}(1-\gamma)}\right)\log\left(\frac{T|\mathcal{S}||\mathcal{A}|}{\delta}\right)\log\left(\frac{1}{\epsilon(1-\gamma)^2}\right),$$

*and step size $0 < \alpha\log\left(\frac{|\mathcal{S}||\mathcal{A}|T}{\delta}\right) < 1$.*

This theorem implies that the convergence rate of our robust Q-learning is as fast as the one of the vanilla Q-learning algorithm in [Li et al., 2020](except the constant $\hat{c}$).

Finally, to show Theorem 3, we only need to show each term in (56) is smaller than $\epsilon$. It can be verified that there exists constants $c_1$, such that if we choose the step size $\alpha = \frac{c_1}{\log\left(\frac{T|\mathcal{S}||\mathcal{A}|}{\delta}\right)}\min\left(\frac{1}{t_{mix}}, \frac{\epsilon^2(1-\gamma)^4}{\gamma^2}\right)$, then $\frac{(1-\rho)^k\|Q_0-Q^*\|_\infty}{1-\gamma} \le \epsilon$ (inequality (51) in [Li et al., 2020]) and $\frac{5\hat{c}\gamma}{1-\gamma}\sqrt{\alpha\log\frac{|\mathcal{S}||\mathcal{A}|T}{\delta}} \le \epsilon$ (by choosing suitable constant $c_1$). Then we have that $\|Q_t - Q^*\|_\infty \le 3\epsilon$. This completes the proof.

## C  Proof of Theorem 4: Approximation of Smoothing Robust Bellman Operator

In this section we prove Theorem 4. First note that for any $x, y \in \mathbb{R}^{|\mathcal{S}|}$,

$$|\text{LSE}(x) - \text{LSE}(y)| \le \sup_{t\in[0,1]} \|\nabla\text{LSE}(tx + (1-t)y)\|_1\|x - y\|_\infty. \tag{57}$$

It can be shown that the gradient of LSE is softmax, i.e.,

$$\frac{\partial \text{LSE}(x)}{\partial x_i} = \frac{e^{\varrho x_i}}{\sum_j e^{\varrho x_j}}. \tag{58}$$

Hence

$$\|\nabla \text{LSE}(z)\|_1 = 1, \forall z \in \mathbb{R}^{|\mathcal{S}|}, \tag{59}$$

which implies that $|\text{LSE}(x) - \text{LSE}(y)| \le \|x - y\|_\infty$. Hence for any $x, y \in \mathbb{R}^{|\mathcal{S}|}$, we have that

$$|\hat{\mathbf{T}}_\pi x(s) - \hat{\mathbf{T}}_\pi y(s)| = \left| \mathbb{E}_A \left[ \gamma(1-R) \sum_{s' \in \mathcal{S}} p^A_{s,s'}(x(s') - y(s')) + \gamma R(\text{LSE}(x) - \text{LSE}(y)) \right] \right|$$
$$\le \gamma(1-R)\|x-y\|_\infty + \gamma R\|x-y\|_\infty$$
$$\le \gamma\|x-y\|_\infty. \tag{60}$$

This means that $\hat{\mathbf{T}}_\pi$ is a contraction, which implies that it has a fixed point.

We then show the limit of the fixed points of $\hat{\mathbf{T}}_\pi$ is the fixed point of $\mathbf{T}_\pi$ Note that $\mathbf{T}_\pi V_1 = V_1$ and $\hat{\mathbf{T}}_\pi V_2 = V_2$, hence

$$\|V_1 - V_2\|_\infty$$
$$= \|\mathbf{T}_\pi V_1 - \hat{\mathbf{T}}_\pi V_2\|_\infty$$
$$\le \|\mathbf{T}_\pi V_1 - \mathbf{T}_\pi V_2\|_\infty + \|\mathbf{T}_\pi V_2 - \hat{\mathbf{T}}_\pi V_2\|_\infty$$
$$= \max_s \left| \mathbb{E}_\pi \left[ \gamma(1-R) \sum_{s'} p^A_{s,s'} V_1(s') + \gamma R \max_{s'} V_1(s') \right. \right.$$
$$\left. \left. - \gamma(1-R) \sum_{s'} p^A_{s,s'} V_2(s') - \gamma R \max_{s'} V_2(s') \right] \right|$$
$$+ \max_s \left| \mathbb{E}_\pi \left[ \gamma R \left( \max_{s'} V_2(s') - \text{LSE}(V_2) \right) \right] \right|$$
$$\le \max_s \mathbb{E}_\pi \left[ \left| \gamma(1-R) \sum_{s'} p^A_{s,s'}(V_1(s') - V_2(s')) \right| + \left| \gamma R \left( \max_{s'} V_1(s') - \max_{s'} V_2(s') \right) \right| \right]$$
$$+ \max_s \left| \mathbb{E}_\pi \left[ \gamma R \left( \max_{s'} V_2(s') - \text{LSE}(V_2) \right) \right] \right|$$
$$\overset{(a)}{\le} \max_s \gamma |V_1(s) - V_2(s)| + \left| \mathbb{E}_\pi \left[ \gamma R \left( \max_{s'} V_2(s') - \text{LSE}(V_2) \right) \right] \right|$$
$$\le \gamma\|V_1 - V_2\|_\infty + \gamma R \frac{\log |\mathcal{S}|}{\varrho}, \tag{61}$$

where $(a)$ is from $|V_1(s') - V_2(s')| \le \max_s |V_1(s) - V_2(s)| = \|V_1 - V_2\|_\infty$ and $|\max_{s'} V_1(s') - \max_{s'} V_2(s')| \le \|V_1 - V_2\|_\infty$, and the last inequality is from $\text{LSE}(V) - \max V = \frac{\log(\sum_s e^{\varrho V(s)}) - \log e^{\varrho \max V}}{\varrho} = \frac{1}{\varrho} \log \frac{\sum_s e^{\varrho V(s)}}{e^{\varrho \max V}} = \frac{1}{\varrho} \log \sum_s e^{\varrho V(s) - \varrho \max V} \le \frac{\log |\mathcal{S}|}{\varrho}$. Hence this completes the proof.

# D   Proof of Theorem 5: Finite-Time Analysis of Robust TDC with Linear Function Approximation

In this section we develop the finite-time analysis of the robust TDC algorithm. In the following proofs, $\|v\|$ denotes the $l_2$ norm if $v$ is a vector; and $\|A\|$ denotes the operator norm if $A$ is a matrix.

For the convenience of proof, we add a projection step to the algorithm, i.e., we let

$$\theta_{t+1} \leftarrow \mathbf{\Pi}_K \left( \theta_t + \alpha \left( \delta_t(\theta_t) \phi_t - \gamma \left( (1-R)\phi_{t+1} + R \sum_{s \in \mathcal{S}} \left( \frac{e^{\varrho V_\theta(s)} \phi_s}{\sum_{j \in \mathcal{S}} e^{\varrho V_\theta(j)}} \right) \right) \phi_t^\top \omega_t \right) \right),$$

$$\omega_{t+1} \leftarrow \mathbf{\Pi}_K \left( \omega_t + \beta(\delta_t(\theta_t) - \phi_t^\top \omega_t)\phi_t \right), \tag{62}$$

for some constant $K$. We note that recently there are several works [Srikant and Ying, 2019, Xu and Liang, 2021, Kaledin et al., 2020] on finite-time analysis of RL algorithms that do not need the projection. However, a direct generalization of their approach does not necessarily work in our case. Specifically, the problem in [Srikant and Ying, 2019] is for one time scale *linear* stochastic approximation. and doesn't need to consider the effect of the $\omega_t$ introduced, also their work highly depends on the bound of the update functions of $\theta_t$ (see inequality (18) in [Srikant and Ying, 2019]). The parameter $\theta_t$ in [Srikant and Ying, 2019] is bounded using itself at a previous timestep by taking advantage of the fact that the update of $\theta$ is linear. However, in our problem, the update is not linear in $\theta$, and our update rule is two time-scale. The approach in [Kaledin et al., 2020] transforms the original two time-scale updates into two asymptotically independent updates via a linear mapping, which is however challenging for our non-linear updates. Some other work, e.g., [Xu and Liang, 2021], gets around this issue by imposing additional assumptions on the function class. Specifically, it is assumed that $V_\theta$ (non-linear function approximation) is bounded for all $\theta$. For the linear function approximation setting considered in this paper, this assumption is equivalent to the assumption of a finite $\theta$, which is guaranteed by the projection step in this paper.

### D.1 Lipschitz Smoothness

In this section, we first show that $\nabla J(\theta)$ is Lipschitz. We begin with an important lemma.

**Lemma 5.** *For any $(s, a, s') \in \mathcal{S} \times \mathcal{A} \times \mathcal{S}$, both $\delta_{s,a,s'}(\theta)$ and $\nabla\delta_{s,a,s'}(\theta)$ are bounded and Lipschitz, i.e., for any $\theta$ and $\theta'$,*

$$|\delta_{s,a,s'}(\theta)| \le c_{\max} + \gamma R(K + \frac{\log|\mathcal{S}|}{\varrho}) + (1+\gamma)K \triangleq C_\delta, \tag{63}$$

$$\|\delta_{s,a,s'}(\theta) - \delta_{s,a,s'}(\theta')\| \le (1+\gamma)\|\theta - \theta'\| \triangleq L_\delta\|\theta - \theta'\|, \tag{64}$$

$$\|\nabla\delta_{s,a,s'}(\theta) - \nabla\delta_{s,a,s'}(\theta')\| \le 2\gamma R\varrho\|\theta - \theta'\| \triangleq L'_\delta\|\theta - \theta'\|. \tag{65}$$

*Proof.* **1. $\delta$ is bounded:**

Recall that

$$\delta_{s,a,s'}(\theta) = c(s,a) + \gamma(1-R)V_\theta(s') + \gamma R\frac{\log(\sum_{j\in\mathcal{S}} e^{\varrho\theta^\top \phi_j})}{\varrho} - V_\theta(s). \tag{66}$$

First we have that

$$|\delta_{s,a,s'}(\theta)| \le c_{\max} + \gamma(1-R)K + \gamma R\frac{\log|\mathcal{S}|e^{K\varrho}}{\varrho} + \gamma RK + K$$

$$= c_{\max} + \gamma R(K + \frac{\log|\mathcal{S}|}{\varrho}) + (1+\gamma)K. \tag{67}$$

**2. $\delta$ is Lipschitz:**

The Lipschitz smoothness of $\delta_{s,a,s'}$ can be showed by finding the bound of $\nabla\delta_{s,a,s'}$. We first recall that

$$\nabla\delta_{s,a,s'}(\theta) = \gamma(1-R)\phi_{s'} + \gamma R\frac{\sum_i e^{\varrho\theta^\top \phi_i}\phi_i}{\sum_j e^{\varrho\theta^\top \phi_j}} - \phi_s. \tag{68}$$

Hence

$$\|\nabla\delta_{s,a,s'}(\theta)\| \le \gamma(1-R) + 1 + \gamma R = 1 + \gamma. \tag{69}$$

**3. $\nabla\delta$ is Lipschitz:**

Finally we need to verify the Lipschitz smoothness of $\nabla\delta_{s,a,s'}(\theta)$, which can be implied from the bound of $\nabla^2\delta_{s,a,s'}(\theta)$. First we have that

$$\nabla^2\delta_{s,a,s'}(\theta) = \gamma R\varrho\frac{\sum_{i,j} e^{\varrho\theta^\top(\phi_i+\phi_j)}\phi_i^\top\phi_i - \sum_{i,j} e^{\varrho\theta^\top(\phi_i+\phi_j)}\phi_i^\top\phi_j}{(\sum_j e^{\varrho\theta^\top \phi_j})^2} \le 2\gamma R\varrho. \tag{70}$$

$\square$

With this lemma, we then show that $\nabla J(\theta)$ is Lipschitz as follows.

**Lemma 6.** *For any $\theta$ and $\theta'$, we have that*

$$\|\nabla J(\theta) - \nabla J(\theta')\| \leq 2\left(\frac{L_\delta^2}{\lambda} + \frac{C_\delta L_\delta'}{\lambda}\right)\|\theta - \theta'\| \triangleq L_J\|\theta - \theta'\|. \tag{71}$$

*Proof.* From Lemma 5, we have that

$$\|\mathbb{E}_{\mu_\pi}[(\nabla\delta_{S,A,S'}(\theta))\phi_S]\| \leq L_\delta \tag{72}$$

and

$$\|\mathbb{E}_{\mu_\pi}[(\nabla\delta_{S,A,S'}(\theta))\phi_S] - \mathbb{E}_{\mu_\pi}[(\nabla\delta_{S,A,S'}(\theta'))\phi_S]\| \leq L_\delta'\|\theta - \theta'\|. \tag{73}$$

Also it is easy to see that

$$\|C^{-1}\mathbb{E}_{\mu_\pi}[\delta_{S,A,S'}(\theta)\phi_S]\| \leq \frac{1}{\lambda}C_\delta, \tag{74}$$

and

$$\|C^{-1}\mathbb{E}_{\mu_\pi}[\delta_{S,A,S'}(\theta)\phi_S] - C^{-1}\mathbb{E}_{\mu_\pi}[\delta_{S,A,S'}(\theta')\phi_S]\| \leq \frac{1}{\lambda}L_\delta\|\theta - \theta'\|. \tag{75}$$

Thus this implies that

$$\|\nabla J(\theta) - \nabla J(\theta')\| \leq 2\left(\frac{L_\delta^2}{\lambda} + \frac{C_\delta L_\delta'}{\lambda}\right)\|\theta - \theta'\|, \tag{76}$$

and hence completes the proof. □

### D.2 Tracking Error

In this section, we study the bound of the tracking error, which is defined as $z_t = \omega_t - \omega(\theta_t)$. First we can rewrite the fast time-scale update in Algorithm 1 as follows:

$$\begin{aligned}
z_{t+1} &= \omega_{t+1} - \omega(\theta_{t+1}) \\
&= \omega_t + \beta(\delta_t(\theta_t) - \phi_t^\top\omega_t)\phi_t - \omega(\theta_{t+1}) \\
&= z_t + \omega(\theta_t) + \beta(\delta_t(\theta_t) - \phi_t^\top\omega_t)\phi_t - \omega(\theta_{t+1}) \\
&= z_t + \omega(\theta_t) + \beta(\delta_t(\theta_t) - \phi_t^\top(z_t + \omega(\theta_t)))\phi_t - \omega(\theta_{t+1}) \\
&= z_t + \omega(\theta_t) + \beta\delta_t(\theta_t)\phi_t - \beta\phi_t^\top z_t\phi_t - \beta\phi_t^\top\omega(\theta_t)\phi_t - \omega(\theta_{t+1}) \\
&= z_t - \beta\phi_t\phi_t^\top z_t + \beta(\delta_t(\theta_t)\phi_t - \phi_t\phi_t^\top\omega(\theta_t)) + \omega(\theta_t) - \omega(\theta_{t+1}). 
\end{aligned} \tag{77}$$

Thus taking the norm of both sides implies that

$$\begin{aligned}
\|z_{t+1}\|^2 &\overset{(a)}{\leq} \|z_t\|^2 + 3\beta^2\|z_t\|^2 + 3\beta^2\|\delta_t(\theta_t)\phi_t - \phi_t\phi_t^\top\omega(\theta_t)\|^2 + 3\|\omega(\theta_t) - \omega(\theta_{t+1})\|^2 \\
&\quad + 2\langle z_t, -\beta\phi_t\phi_t^\top z_t\rangle + 2\langle z_t, \beta(\delta_t(\theta_t)\phi_t - \phi_t\phi_t^\top\omega(\theta_t))\rangle + 2\langle z_t, \omega(\theta_t) - \omega(\theta_{t+1})\rangle \\
&= \|z_t\|^2 - 2\beta z_t^\top C z_t + 3\beta^2\|z_t\|^2 + 3\beta^2\|\delta_t(\theta_t)\phi_t - \phi_t\phi_t^\top\omega(\theta_t)\|^2 + 3\|\omega(\theta_t) - \omega(\theta_{t+1})\|^2 \\
&\quad + 2\beta\langle z_t, (C - \phi_t\phi_t^\top)z_t\rangle + 2\langle z_t, \beta(\delta_t(\theta_t)\phi_t - \phi_t\phi_t^\top\omega(\theta_t))\rangle + 2\langle z_t, \omega(\theta_t) - \omega(\theta_{t+1})\rangle \\
&\overset{(b)}{\leq} (1 + 3\beta^2 - 2\beta\lambda)\|z_t\|^2 + \beta^2 C_1 + 2\beta\langle z_t, (C - \phi_t\phi_t^\top)z_t\rangle + 2\langle z_t, \omega(\theta_t) - \omega(\theta_{t+1})\rangle \\
&\quad + 2\langle z_t, \beta(\delta_t(\theta_t)\phi_t - \phi_t\phi_t^\top\omega(\theta_t))\rangle,
\end{aligned} \tag{78}$$

where $(a)$ is from $\|x + y + z\|^2 \leq 3\|x\|^2 + 3\|y\|^2 + 3\|z\|^2$ for any $x, y, z \in \mathbb{R}^N$, $(b)$ is from $z_t^\top C z_t \geq \lambda\|z_t\|^2$, and $C_1 = 3\left(C_\delta + \frac{C_\delta}{\lambda}\right)^2 + 3\left(C_\delta + (1 + 2R\varrho K)\frac{C_\delta}{\lambda}\right)^2$ is the upper bound of $3\|\delta_t(\theta_t)\phi_t - \phi_t\phi_t^\top\omega(\theta_t)\|^2 + \frac{3}{\beta^2}\|\omega(\theta_t) - \omega(\theta_{t+1})\|^2$.

Taking expectation on both sides and applying recursively (78), we obtain that

$$\mathbb{E}[\|z_{t+1}\|^2] \leq q^{t+1}\|z_0\|^2 + 2\sum_{j=0}^t q^{t-j}\beta\mathbb{E}[f(z_j, O_j)] + 2\sum_{j=0}^t q^{t-j}\beta\mathbb{E}[g(z_j, \theta_j, O_j)]$$

$$+ 2\sum_{j=0}^{t} q^{t-j}\langle z_j, \omega(\theta_j) - \omega(\theta_{j+1})\rangle + \beta^2 C_1 \sum_{j=0}^{t} q^{t-j}, \tag{79}$$

where

$$q \triangleq 1 + 3\beta^2 - 2\beta\lambda,$$
$$f(z_j, O_j) \triangleq \langle z_j, (C - \phi_j\phi_j^\top)z_j\rangle,$$
$$g(z_j, \theta_j, O_j) \triangleq \langle z_j, \delta_j(\theta_j)\phi_j - \phi_j\phi_j^\top\omega(\theta_j)\rangle. \tag{80}$$

To simplify notations, let

$$\theta_{t+1} \leftarrow \theta_t + \alpha G_t(\theta_t, \omega_t), \tag{81}$$
$$\omega_{t+1} \leftarrow \omega_t + \beta H_t(\theta_t, \omega_t), \tag{82}$$

where $G_t(\theta, \omega) = \delta_t(\theta)\phi_t - \gamma\left((1-R)\phi_{t+1} + R\frac{\sum_i e^{\theta\theta^\top\phi_i\phi_i}}{\sum_j e^{\theta\theta^\top\phi_j}}\right)\phi_t^\top\omega$, and $H_t(\theta, \omega) = (\delta_t(\theta) - \phi_t^\top\omega_t)\phi_t$.

We have

$$\|G_t(\theta, \omega)\| \le C_\delta + K\gamma \triangleq C_G. \tag{83}$$

The upper bound of $H_t(\theta, \omega)$ is straightforward:

$$\|H_t(\theta, \omega)\| \le C_\delta + K \triangleq C_H. \tag{84}$$

With these two bounds we can then find the upper bound of the update of tracking error:

$$\begin{aligned}
\|z_{t+1} - z_t\| &\le \|H_t(\theta_t, \omega_t)\| + \|\omega(\theta_{t+1}) - \omega(\theta_t)\| \\
&\overset{(a)}{\le} \beta C_H + \alpha\frac{C_\delta}{\lambda}\|G_t(\theta_t, \omega_t)\| \\
&\le \beta C_H + \alpha\frac{C_\delta C_G}{\lambda},
\end{aligned} \tag{85}$$

where $(a)$ is from the Lipschitz of $\omega(\theta)$: $\|\omega(\theta_{t+1}) - \omega(\theta_t)\| \le \frac{L_\delta}{\lambda}\|\theta_{t+1} - \theta_t\| \le \frac{\alpha L_\delta}{\lambda}\|G_t(\theta_t, \omega_t)\|$. Then for the Lipschitz smoothness of function $g$ in (80), it is straightforward to see that

$$\begin{aligned}
&|g(\theta, z, O_t) - g(\theta', z', O_t)| \\
&= \langle z, \delta_j(\theta)\phi_j - \phi_j\phi_j^\top\omega(\theta)\rangle - \langle z', \delta_j(\theta')\phi_j - \phi_j\phi_j^\top\omega(\theta')\rangle \\
&= \langle z, \delta_j(\theta)\phi_j - \phi_j\phi_j^\top\omega(\theta)\rangle - \langle z, \delta_j(\theta')\phi_j - \phi_j\phi_j^\top\omega(\theta')\rangle \\
&\quad + \langle z, \delta_j(\theta')\phi_j - \phi_j\phi_j^\top\omega(\theta')\rangle - \langle z', \delta_j(\theta')\phi_j - \phi_j\phi_j^\top\omega(\theta')\rangle \\
&\le K_z L_\delta\left(1 + \frac{1}{\lambda}\right)\|\theta - \theta'\| + C_\delta\left(1 + \frac{1}{\lambda}\right)\|z - z'\|,
\end{aligned} \tag{86}$$

where $K_z \triangleq K + \frac{C_\delta}{\lambda}$ being a rough bound on the track error. Also it can be shown that

$$\begin{aligned}
|f(z, O_t) - f(z', O_t)| &= \langle z, (C - \phi_t\phi_t^\top)z\rangle - \langle z', (C - \phi_t\phi_t^\top)z'\rangle \\
&= \langle z, (C - \phi_t\phi_t^\top)z\rangle - \langle z, (C - \phi_t\phi_t^\top)z'\rangle \\
&\quad + \langle z, (C - \phi_t\phi_t^\top)z'\rangle - \langle z', (C - \phi_t\phi_t^\top)z'\rangle \\
&\le 4K_z\|z - z'\|.
\end{aligned} \tag{87}$$

It is easy to see that

$$\|G_i(\theta, \omega_1) - G_i(\theta, \omega_2)\| \le (\gamma + 2\gamma R\varrho K)\|\omega_1 - \omega_2\|. \tag{88}$$

With these bounds and Lipschitz constants, the following two lemmas can be proved using the similar method of decoupling the Markovian noise in [Wang and Zou, 2020, Bhandari et al., 2018, Zou et al., 2019].

**Lemma 7.** *Define* $\tau_\beta = \min\{k : m\rho^k \leq \beta\}$. *If* $t < \tau_\beta$, *then*

$$\mathbb{E}[f(z_t, O_t)] \leq 4K_z^2; \tag{89}$$

*and if* $t \geq \tau_\beta$, *then*

$$\mathbb{E}[f(z_t, O_t)] \leq m_f\beta + m'_f\tau_\beta\beta, \tag{90}$$

*where* $m_f = 8K_z^2$ *and* $m'_f = 8K_z\left(C_H + \frac{C_G C_\delta}{\lambda}\right)$.

A similar result on $\mathbb{E}[g(\theta_t, z_t, O_t)]$ can also be implied:

**Lemma 8.** *If* $t < \tau_\beta$, *then*

$$\mathbb{E}[g(\theta_t, z_t, O_t)] \leq 2K_z\left(1 + \frac{1}{\lambda}\right)C_\delta; \tag{91}$$

*and if* $t \geq \tau_\beta$, *then*

$$\mathbb{E}[g(\theta_t, z_t, O_t)] \leq m_g\beta + m'_g\tau_\beta\beta, \tag{92}$$

*where* $m_g = 4K_z\left(1 + \frac{1}{\lambda}\right)C_\delta$ *and* $m'_g = 4K_z L_\delta C_G\left(1 + \frac{1}{\lambda}\right) + C_\delta\left(1 + \frac{1}{\lambda}\right)\left(C_H + \frac{C_G C_\delta}{\lambda}\right)$.

One more lemma is needed to bound the tracking error.

**Lemma 9.** *Define* $h(\theta, z, O_t) = \left\langle z, -\nabla\omega(\theta)\left(G_t(\theta, \omega(\theta)) + \frac{\nabla J(\theta)}{2}\right)\right\rangle$, *then if* $t < \tau_\beta$,

$$\mathbb{E}[h(\theta_t, z_t, O_t)] \leq K_z C_h; \tag{93}$$

*and if* $t \geq \tau_\beta$,

$$\mathbb{E}[h(\theta_t, z_t, O_t)] \leq m_h\beta + m'_h\tau_\beta\beta, \tag{94}$$

*where* $m_h = 2K_z C_h$ *and* $m'_h = C_h\left(C_H + \frac{C_\delta C_G}{\lambda}\right) + K_z L_h C_G$.

*Proof.* First we show the Lipschitz smoothness of $h$ as follows. For any $\theta, \theta', z$ and $z'$, we have that

$$h(\theta, z, O_t) - h(\theta', z', O_t)$$
$$= \left\langle z, -\nabla\omega(\theta)\left(G_t(\theta, \omega(\theta)) + \frac{\nabla J(\theta)}{2}\right)\right\rangle - \left\langle z', -\nabla\omega(\theta')\left(G_t(\theta', \omega(\theta')) + \frac{\nabla J(\theta')}{2}\right)\right\rangle$$
$$= \left\langle z, -\nabla\omega(\theta)\left(G_t(\theta, \omega(\theta)) + \frac{\nabla J(\theta)}{2}\right)\right\rangle - \left\langle z', -\nabla\omega(\theta)\left(G_t(\theta, \omega(\theta)) + \frac{\nabla J(\theta)}{2}\right)\right\rangle$$
$$+ \left\langle z', -\nabla\omega(\theta)\left(G_t(\theta, \omega(\theta)) + \frac{\nabla J(\theta)}{2}\right)\right\rangle - \left\langle z', -\nabla\omega(\theta')\left(G_t(\theta', \omega(\theta')) + \frac{\nabla J(\theta')}{2}\right)\right\rangle. \tag{95}$$

We note that

$$\left\|-\nabla\omega(\theta)\left(G_t(\theta, \omega(\theta)) + \frac{\nabla J(\theta)}{2}\right)\right\|$$
$$\leq \frac{L_\delta}{\lambda}\left(C_\delta + \gamma(1 - R) + 2\varrho K\gamma R\frac{C_\delta}{\lambda} + \frac{2L_\delta C_\delta}{\lambda}\right) \triangleq C_h, \tag{96}$$

and

$$\left\|-\nabla\omega(\theta)\left(G_t(\theta, \omega(\theta)) + \frac{\nabla J(\theta)}{2}\right) + \nabla\omega(\theta')\left(G_t(\theta', \omega(\theta')) + \frac{\nabla J(\theta')}{2}\right)\right\|$$
$$\leq \left(\frac{L'_\delta}{L_\delta}C_h + \frac{L_\delta L_{G^*}}{\lambda} + \frac{L_\delta L_J}{2\lambda}\right)\|\theta - \theta'\| \triangleq L_h\|\theta - \theta'\|. \tag{97}$$

Hence we have that

$$h(\theta, z, O_t) - h(\theta', z', O_t) \leq C_h\|z - z'\| + K_z L_h\|\theta - \theta'\|. \tag{98}$$

We have shown before in (85) that

$$\|z_{t+1} - z_t\| \le \beta C_H + \alpha \frac{C_\delta C_G}{\lambda}. \tag{99}$$

Hence, we have that

$$|h(\theta_t, z_t, O_t) - h(\theta_{t-\tau}, z_{t-\tau}, O_t)| \le C_h \left( \beta C_H + \alpha \frac{C_\delta C_G}{\lambda} \right) \tau + K_z L_h C_G \tau \alpha. \tag{100}$$

Define an independent random variable $\hat{O} = (\hat{S}, \hat{A}, \hat{S}') \sim \mu_\pi \times \mathsf{P}(\cdot | \hat{S}, \hat{A})$, then we have

$$\mathbb{E}_{\hat{O}}[h(\theta, z, \hat{O})] = 0 \tag{101}$$

for any $\theta$ and $z$. Thus by uniform ergodicity, we have that

$$\mathbb{E}[h(\theta_{t-\tau}, z_{t-\tau}, O_t)] \le \mathbb{E}[h(\theta_{t-\tau}, z_{t-\tau}, O_t)] - \mathbb{E}_{\hat{O}}[h(\theta_t, z_t, \hat{O})] \le 2K_z C_h m \rho^\tau. \tag{102}$$

Then if $t \le \tau_\beta$, we have the straightforward bound

$$\mathbb{E}[h(\theta_t, z_t, O_t)] \le K_z C_h; \tag{103}$$

and if $t > \tau_\beta$, we have that

$$\mathbb{E}[h(\theta_t, z_t, O_t)] \le \mathbb{E}[h(\theta_{t-\tau_\beta}, z_{t-\tau_\beta}, O_t)] + C_h \left( \beta C_H + \alpha \frac{C_\delta C_G}{\lambda} \right) \tau_\beta + K_z L_h C_G \tau_\beta \alpha$$

$$\le 2K_z C_h m \rho^{\tau_\beta} + C_h \left( \beta C_H + \alpha \frac{C_\delta C_G}{\lambda} \right) \tau_\beta + K_z L_h C_G \tau_\beta \alpha$$

$$\triangleq m_h \beta + m'_h \tau_\beta \beta, \tag{104}$$

where $m_h = 2K_z C_h$ and $m'_h = C_h \left( C_H + \frac{C_\delta C_G}{\lambda} \right) + K_z L_h C_G$. This completes the proof. $\qquad\square$

Now we bound the tracking error in (79). We first rewrite it as

$$\mathbb{E}[\|z_{t+1}\|^2] \le q^{t+1} \|z_0\|^2 + \underbrace{2\sum_{j=0}^{t} q^{t-j} \beta \mathbb{E}[f(z_j, O_j)]}_{A_t} + \underbrace{2\sum_{j=0}^{t} q^{t-j} \beta \mathbb{E}[g(z_j, \theta_j, O_j)]}_{B_t}$$

$$+ \underbrace{2\sum_{j=0}^{t} q^{t-j} \langle z_j, \omega(\theta_j) - \omega(\theta_{j+1}) \rangle}_{C_t} + \beta^2 C_1 \sum_{j=0}^{t} q^{t-j}. \tag{105}$$

The second term $A_t$ can be bounded as follows:

$$A_t = 2\sum_{j=0}^{t} q^{t-j} \beta \mathbb{E}[f(z_j, O_j)]$$

$$= 2\sum_{j=0}^{\tau_\beta - 1} q^{t-j} \beta \mathbb{E}[f(z_j, O_j)] + 2\sum_{j=\tau_\beta}^{t} q^{t-j} \beta \mathbb{E}[\hat{f}(z_j, O_j)]$$

$$\le 8\sum_{j=0}^{\tau_\beta - 1} q^{t-j} K_z \beta + 2\sum_{j=\tau_\beta}^{t} q^{t-j} \beta (m_f \beta + m'_f \tau_\beta \beta)$$

$$\le 16 K_z \beta \frac{q^{t+1-\tau_\beta}}{1-q} + 2\beta (m_f \beta + m'_f \tau_\beta \beta) \frac{1 - q^{t-\tau_\beta+1}}{1-q}. \tag{106}$$

Similarly, we have that

$$B_t \le 4 K_z \beta \left( 1 + \frac{1}{\lambda} \right) C_\delta \frac{q^{t+1-\tau_\beta}}{1-q} + 2\beta (m_g \beta + m'_g \tau_\beta \beta) \frac{1 - q^{t-\tau_\beta+1}}{1-q}. \tag{107}$$

For $C_t$, we first note that

$$\mathbb{E}\left[\langle z_i, \omega(\theta_i) - \omega(\theta_{i+1})\rangle\right]$$

$$\stackrel{(a)}{=} \mathbb{E}\left[\langle z_i, \nabla\omega(\theta_i)(\theta_i - \theta_{i+1}) + R_2\rangle\right]$$

$$= \mathbb{E}\left[\langle z_i, -\alpha\nabla\omega(\theta_i)G_i(\theta_i,\omega_i) + R_2\rangle\right]$$

$$= \mathbb{E}\left[\left\langle z_i, -\alpha\nabla\omega(\theta_i)\left(G_i(\theta_i,\omega_i) - G_i(\theta_i,\omega(\theta_i)) + G_i(\theta_i,\omega(\theta_i)) + \frac{\nabla J(\theta_i)}{2} - \frac{\nabla J(\theta_i)}{2}\right)\right.\right.$$

$$\left.\left. + R_2\right\rangle\right]$$

$$= \underbrace{\mathbb{E}\left[\left\langle z_i, -\alpha\nabla\omega(\theta_i)\left(G_i(\theta_i,\omega(\theta_i)) + \frac{\nabla J(\theta_i)}{2}\right)\right\rangle\right]}_{(b)}$$

$$+ \underbrace{\mathbb{E}\left[\left\langle z_i, -\alpha\nabla\omega(\theta_i)\left(G_i(\theta_i,\omega_i) - G_i(\theta_i,\omega(\theta_i)) - \frac{\nabla J(\theta_i)}{2}\right) + R_2\right\rangle\right]}_{(c)}, \tag{108}$$

where $(a)$ follows from the Taylor expansion, and $R_2$ is the remaining term with norm $\|R_2\| = \mathcal{O}(\alpha^2)$. Term $(b)$ can be bounded using Lemma 9, where

$$\mathbb{E}\left[\left\langle z_i, -\alpha\nabla\omega(\theta_i)\left(G_i(\theta_i,\omega(\theta_i)) + \frac{\nabla J(\theta_i)}{2}\right)\right\rangle\right] = \alpha\mathbb{E}\left[h(\theta_i,z_i,O_i)\right]. \tag{109}$$

Term $(c)$ can be bounded as follows.

$$\left\langle z_i, -\alpha\nabla\omega(\theta_i)\left(G_i(\theta_i,\omega_i) - G_i(\theta_i,\omega(\theta_i)) - \frac{\nabla J(\theta_i)}{2}\right) + R_2\right\rangle$$

$$\stackrel{(d)}{\leq} \frac{\lambda\beta}{8}\|z_i\|^2 + \frac{2}{\lambda\beta}\left\|\alpha\nabla\omega(\theta_i)\left(G_i(\theta_i,\omega_i) - G_i(\theta_i,\omega(\theta_i)) - \frac{\nabla J(\theta_i)}{2}\right) + R_2\right\|^2$$

$$\leq \frac{\lambda\beta}{8}\|z_i\|^2$$

$$+ \frac{6}{\lambda\beta}\left(\|\alpha\nabla\omega(\theta_i)(G_i(\theta_i,\omega_i) - G_i(\theta_i,\omega(\theta_i)))\|^2 + \left\|\alpha\nabla\omega(\theta_i)\frac{\nabla J(\theta_i)}{2}\right\|^2 + \|R_2\|^2\right)$$

$$\leq \frac{\lambda\beta}{8}\|z_i\|^2 + \frac{6\alpha^2}{\lambda\beta}\frac{L_\delta^2}{\lambda^2}(\gamma + 2\gamma R\varrho K)^2\|z_i\|^2 + \frac{3\alpha^2}{2\lambda\beta}\frac{L_\delta^2}{\lambda^2}\|\nabla J(\theta_i)\|^2 + \frac{6}{\lambda\beta}\|R_2\|^2. \tag{110}$$

where $(d)$ is from $\langle x,y\rangle \leq \frac{\lambda\beta}{8}\|x\|^2 + \frac{2}{\lambda\beta}\|y\|^2$ for any $x,y \in \mathbb{R}^N$ and the fact that $\|G_i(\theta,\omega_1) - G_i(\theta,\omega_2)\| \leq (\gamma + 2\gamma\varrho RK)\|\omega_1 - \omega_2\|$ for any $\|\theta\| \leq R$ and $\omega_1,\omega_2$, which is from (88).

Finally the term $C_t$ can be bounded as follows.

$$C_t = 2\sum_{j=0}^{t} q^{t-j}\langle z_j, \omega(\theta_j) - \omega(\theta_{j+1})\rangle$$

$$= 2\sum_{j=0}^{t} q^{t-j}\alpha\mathbb{E}[h(\theta_j,z_j,O_j)]$$

$$+ 2\sum_{j=0}^{t} q^{t-j}\left(\frac{\lambda\beta}{8}\|z_i\|^2 + \frac{6\alpha^2}{\lambda\beta}\frac{L_\delta^2}{\lambda^2}(\gamma + 2\gamma R\varrho K)^2\|z_i\|^2 + \frac{3\alpha^2}{2\lambda\beta}\frac{L_\delta^2}{\lambda^2}\|\nabla J(\theta_i)\|^2 + \frac{6}{\lambda\beta}\|R_2\|^2\right)$$

$$\triangleq 2\sum_{j=0}^{t} q^{t-j}\alpha\mathbb{E}[h(\theta_j,z_j,O_j)] + M_t, \tag{111}$$

where $M_t = 2\sum_{j=0}^{t} q^{t-j}\left(\frac{\lambda\beta}{8}\|z_i\|^2 + \frac{6\alpha^2}{\lambda\beta}\frac{L_\delta^2}{\lambda^2}(\gamma + 2\gamma R\varrho K)^2\|z_i\|^2 + \frac{3\alpha^2}{2\lambda\beta}\frac{L_\delta^2}{\lambda^2}\|\nabla J(\theta_i)\|^2 + \frac{6}{\lambda\beta}\|R_2\|^2\right)$.
From Lemma 9, we have that

$$2\sum_{j=0}^{t} q^{t-j}\alpha\mathbb{E}[h(\theta_j, z_j, O_j)]$$

$$\leq 2\alpha\left(\sum_{j=0}^{\tau_\beta-1} q^{t-j}\mathbb{E}[h(\theta_j, z_j, O_j)] + \sum_{j=\tau_\beta}^{t} q^{t-j}\mathbb{E}[h(\theta_j, z_j, O_j)]\right)$$

$$\leq 4K_z C_h\alpha\sum_{j=0}^{\tau_\beta-1} q^{t-j} + 2\alpha(m_h\beta + m_h'\tau_\beta\beta)\sum_{j=\tau_\beta}^{t} q^{t-j}$$

$$= 4K_z C_h\alpha\frac{q^{t+1-\tau_\beta}}{1-q} + 2\alpha(m_h\beta + m_h'\tau_\beta\beta)\frac{1 - q^{t-\tau_\beta+1}}{1-q}, \tag{112}$$

and this implies that

$$C_t \leq 4K_z C_h\alpha\frac{q^{t+1-\tau_\beta}}{1-q} + 2\alpha(m_h\beta + m_h'\tau_\beta\beta)\frac{1 - q^{t-\tau_\beta+1}}{1-q} + M_t. \tag{113}$$

Now we plug the bounds on $A_t$, $B_t$ and $C_t$ in (79), we have that

$$\mathbb{E}[\|z_{t+1}\|^2]$$

$$\leq q^{t+1}\|z_0\|^2 + \beta^2 C_1\frac{1 - q^{t+1}}{1-q} + \left(16K_z\beta + 4K_z C_\delta\beta\left(1 + \frac{1}{\lambda}\right) + 4K_z C_h\alpha\right)\frac{q^{t+1-\tau_\beta}}{1-q}$$

$$+ \left(2\beta(m_f\beta + m_f'\tau_\beta\beta) + 2\beta(m_g\beta + m_g'\tau_\beta\beta) + 2\alpha(m_h\beta + m_h'\tau_\beta\beta)\right)\frac{1 - q^{t-\tau_\beta+1}}{1-q} + M_t$$

$$\leq q^{t+1}\|z_0\|^2 + \beta^2 C_1\frac{1 - q^{t+1}}{1-q} + C_z\beta\frac{q^{t+1-\tau_\beta}}{1-q} + \beta(m_z\beta + m_z'\tau_\beta\beta)\frac{1 - q^{t-\tau_\beta+1}}{1-q} + M_t, \tag{114}$$

where $C_z = 16K_z + 4K_z C_\delta\left(1 + \frac{1}{\lambda}\right) + 4K_z C_h\frac{\alpha}{\beta}$, $m_z = 2m_f + 2m_g + 2\frac{\alpha}{\beta}m_h$ and $m_z' = 2m_f' + 2m_g' + \frac{2\alpha}{\beta}m_h'$. Note that $q = 1 + 3\beta^2 - 2\beta\lambda \triangleq 1 - u\beta \leq e^{-u\beta}$, where $u = 2\lambda - 3\beta$. Hence it implies that

$$\frac{\sum_{t=0}^{T-1}\mathbb{E}[\|z_t\|^2]}{T}$$

$$\leq \frac{1}{T}\left(\frac{\|z_0\|^2}{1 - e^{-u\beta}} + \beta^2 C_1\frac{T}{u\beta} + 4K_z^2\tau_\beta\right.$$

$$\left.+ \sum_{t=\tau_\beta-1}^{T-1}\left(C_z\beta\frac{q^{t+1-\tau_\beta}}{u\beta} + \beta(m_z\beta + m_z'\tau_\beta\beta)\frac{1 - q^{t-\tau_\beta+1}}{u\beta} + M_t\right)\right)$$

$$\leq \frac{1}{T}\left(\frac{\|z_0\|^2}{1 - e^{-u\beta}} + \beta^2 C_1\frac{T}{u\beta} + 4K_z^2\tau_\beta\right.$$

$$\left.+ c_z\beta\frac{\sum_{t=0}^{T-1} e^{-ut\beta}}{u\beta} + \beta(m_z\beta + m_z'\tau_\beta\beta)\frac{T}{u\beta} + \sum_{t=0}^{T-1} M_t\right)$$

$$\leq \frac{1}{T}\left(\frac{\|z_0\|^2}{1 - e^{-u\beta}} + \beta^2 C_1\frac{T}{u\beta} + 4K_z^2\tau_\beta + c_z\beta\frac{1}{(u\beta)(1 - e^{-u\beta})} + \beta(m_z\beta + m_z'\tau_\beta\beta)\frac{T}{u\beta}\right.$$

$$\left.+ \sum_{t=0}^{T-1} M_t\right)$$

$$= \frac{1}{T}\left(\frac{\|z_0\|^2}{1 - e^{-u\beta}} + \beta C_1\frac{T}{u} + 4K_z^2\tau_\beta + \frac{c_z}{u(1 - e^{-u\beta})} + (m_z\beta + m_z'\tau_\beta\beta)\frac{T}{u} + \sum_{t=0}^{T-1} M_t\right)$$

$$\leq \frac{\|z_0\|^2}{T(1-e^{-u\beta})} + \beta\frac{C_1}{u} + 4K_z^2\frac{\tau_\beta}{T} + \frac{c_z}{u(1-e^{-u\beta})T} + (m_z\beta + m_z'\tau_\beta\beta)\frac{1}{u} + \frac{\sum_{t=0}^{T-1}M_t}{T}$$

$$\triangleq Q_T + \frac{\sum_{t=0}^{T-1}M_t}{T}$$

$$= \mathcal{O}\left(\frac{1}{T\beta} + \beta\tau_\beta + \frac{\tau_\beta}{T} + \frac{\sum_{t=0}^{T-1}M_t}{T}\right), \tag{115}$$

where $Q_T = \frac{\|z_0\|^2}{T(1-e^{-u\beta})} + \beta\frac{C_1}{u} + 4K_z^2\frac{\tau_\beta}{T} + \frac{c_z}{u(1-e^{-u\beta})T} + (m_z\beta + m_z'\tau_\beta\beta)\frac{1}{u}$.

We then compute $\sum_{t=0}^{T-1}M_t$. Recall that $M_t = 2\sum_{j=0}^{t}q^{t-j}\left(\frac{\lambda\beta}{8}\|z_i\|^2 + \frac{6\alpha^2}{\lambda\beta}\frac{L_\delta^2}{\lambda^2}(\gamma + 2\gamma R\varrho K)^2\|z_i\|^2 + \frac{3\alpha^2}{2\lambda\beta}\frac{L_\delta^2}{\lambda^2}\|\nabla J(\theta_i)\|^2 + \frac{6}{\lambda\beta}\|R_2\|^2\right)$. From double sum trick, i.e., $\sum_{t=0}^{T-1}\sum_{i=0}^{t}e^{-u(t-i)\beta}x_i \leq \frac{1}{1-e^{-u\beta}}\sum_{t=0}^{T-1}x_t$ for any $x_t \geq 0$, we have that

$$\sum_{t=0}^{T-1}M_t \leq \frac{2}{1-e^{-u\beta}}\left(\frac{\lambda\beta}{8} + \frac{6\alpha^2}{\lambda\beta}\frac{L_\delta^2}{\lambda^2}(\gamma + 2\gamma R\varrho K)^2\right)\sum_{t=0}^{T-1}\mathbb{E}[\|z_t\|^2]$$

$$+ \frac{2}{1-e^{-u\beta}}\frac{3\alpha^2}{2\lambda\beta}\frac{L_\delta^2}{\lambda^2}\sum_{t=0}^{T-1}\mathbb{E}[\|\nabla J(\theta_t)\|^2] + \frac{6}{\lambda\beta}\frac{2}{1-e^{-u\beta}}\|R_2\|^2 T. \tag{116}$$

Note that $1 - e^{-u\beta} = \mathcal{O}(\beta)$, thus we can choose $\alpha$ and $\beta$ such that $\frac{2}{1-e^{-u\beta}}\left(\frac{\lambda\beta}{8} + \frac{6\alpha^2}{\lambda\beta}\frac{L_\delta^2}{\lambda^2}(\gamma + 2\gamma R\varrho K)^2\right) \leq \frac{1}{2}$, then by plugging $\sum_{t=0}^{T-1}M_t$ in (115) we have that

$$\frac{1}{2}\frac{\sum_{t=0}^{T-1}\mathbb{E}[\|z_t\|^2]}{T} \leq Q_T + \frac{2}{1-e^{-u\beta}}\frac{3\alpha^2}{2\lambda\beta}\frac{L_\delta^2}{\lambda^2}\frac{\sum_{t=0}^{T-1}\mathbb{E}[\|\nabla J(\theta_t)\|^2]}{T} + \frac{6}{\lambda\beta}\frac{2}{1-e^{-u\beta}}\|R_2\|^2, \tag{117}$$

and this implies that

$$\frac{\sum_{t=0}^{T-1}\mathbb{E}[\|z_t\|^2]}{T} \leq 2Q_T + \frac{2}{1-e^{-u\beta}}\frac{3\alpha^2}{\lambda\beta}\frac{L_\delta^2}{\lambda^2}\frac{\sum_{t=0}^{T-1}\mathbb{E}[\|\nabla J(\theta_t)\|^2]}{T} + \frac{6}{\lambda\beta}\frac{4}{1-e^{-u\beta}}\|R_2\|^2$$

$$= \mathcal{O}\left(\frac{1}{T\beta} + \beta\tau_\beta + \frac{\alpha^2}{\beta^2}\frac{\sum_{t=0}^{T-1}\mathbb{E}[\|\nabla J(\theta_t)\|^2]}{T}\right), \tag{118}$$

which completes the development of error bound on the tracking error.

### D.3 Finite-Time Error Bound

Now with the tracking error in (118), we derive the finite-time error of the robust TDC. From Lemma 6 and Taylor expansion, we have that

$$J(\theta_{t+1}) \leq J(\theta_t) + \langle\nabla J(\theta_t), \theta_{t+1} - \theta_t\rangle + \frac{L_J}{2}\|\theta_{t+1} - \theta_t\|^2$$

$$= J(\theta_t) + \alpha\langle\nabla J(\theta_t), G_t(\theta_t, \omega_t)\rangle + \frac{L_J}{2}\alpha^2\|G_t(\theta_t, \omega_t)\|^2$$

$$= J(\theta_t) - \alpha\left\langle\nabla J(\theta_t), -G_t(\theta_t, \omega_t) - \frac{\nabla J(\theta_t)}{2} + G_t(\theta_t, \omega(\theta_t)) - G_t(\theta_t, \omega(\theta_t))\right\rangle$$

$$- \frac{\alpha}{2}\|\nabla J(\theta_t)\|^2 + \frac{L_J}{2}\alpha^2\|G_t(\theta_t, \omega_t)\|^2$$

$$= J(\theta_t) - \alpha\langle\nabla J(\theta_t), -G_t(\theta_t, \omega_t) + G_t(\theta_t, \omega(\theta_t))\rangle$$

$$+ \alpha\left\langle\nabla J(\theta_t), \frac{\nabla J(\theta_t)}{2} + G_t(\theta_t, \omega(\theta_t))\right\rangle - \frac{\alpha}{2}\|\nabla J(\theta_t)\|^2 + \frac{L_J}{2}\alpha^2\|G_t(\theta_t, \omega_t)\|^2$$

$$\leq J(\theta_t) + \alpha\|\nabla J(\theta_t)\|(\gamma + 2\gamma RK\varrho)\|\omega(\theta_t) - \omega_t\| - \frac{\alpha}{2}\|\nabla J(\theta_t)\|^2$$

$$+ \alpha \left\langle \nabla J(\theta_t), \frac{\nabla J(\theta_t)}{2} + G_t(\theta_t, \omega(\theta_t)) \right\rangle + \frac{L_J}{2} \alpha^2 ||G_t(\theta_t, \omega_t)||^2. \tag{119}$$

By taking expectation on both sides and summing up from 0 to $T-1$, we have that

$$\sum_{t=0}^{T-1} \frac{\alpha}{2} \mathbb{E}[||\nabla J(\theta_t)||^2]$$

$$\leq J(\theta_0) - J(\theta_T) + \alpha(\gamma + 2\gamma RK\varrho) \sqrt{\sum_{t=0}^{T-1} \mathbb{E}[||\nabla J(\theta_t)||^2]} \sqrt{\sum_{t=0}^{T-1} \mathbb{E}[||z_t||^2]}$$

$$+ \sum_{t=0}^{T-1} \alpha \mathbb{E} \left[ \left\langle \nabla J(\theta_t), \frac{\nabla J(\theta_t)}{2} + G_t(\theta_t, \omega(\theta_t)) \right\rangle \right] + \frac{L_J}{2} \sum_{t=0}^{T-1} \alpha^2 \mathbb{E}[||G_t(\theta_t, \omega_t)||^2], \tag{120}$$

which follows from the Cauchy-Schwartz inequality: $\sum_{t=0}^{T-1} \mathbb{E}[||\nabla J(\theta_t)|| \, ||z_t||] \leq \sum_{t=0}^{T-1} \sqrt{\mathbb{E}[||\nabla J(\theta_t)||^2] \mathbb{E}[||z_t||^2]} \leq \sqrt{\sum_{t=0}^{T-1} \mathbb{E}[||\nabla J(\theta_t)||^2]} \sqrt{\sum_{t=0}^{T-1} \mathbb{E}[||z_t||^2]}$. To bound the Markovian noise term, i.e., $\left\langle \nabla J(\theta), \frac{\nabla J(\theta)}{2} + G_t(\theta, \omega(\theta)) \right\rangle$, we first need some bounds and smoothness conditions. It can be shown that

$$||G_t(\theta, \omega(\theta))|| \leq C_\delta + \frac{C_\delta}{\lambda}(\gamma + 2\varrho K\gamma R) \triangleq C_{G*}, \tag{121}$$

$$||G_t(\theta, \omega(\theta)) - G_t(\theta', \omega(\theta'))|| \leq \left( L_\delta + \frac{L_\delta}{\lambda}(\gamma + 2\gamma R\varrho K) + \frac{C_\delta}{\lambda} L_\delta' \right) ||\theta - \theta'|| \triangleq L_{G*} ||\theta - \theta'||. \tag{122}$$

**Lemma 10.** *Define* $\zeta(\theta, O_t) \triangleq \left\langle \nabla J(\theta), \frac{\nabla J(\theta)}{2} + G_t(\theta, \omega(\theta)) \right\rangle$, *and let* $\tau_\alpha \triangleq \min\left\{ k : m\rho^k \leq \alpha \right\}$. *If* $t < \tau_\alpha$, *then*

$$\mathbb{E}[\zeta(\theta_t, O_t)] \leq \frac{C_\delta L_\delta}{\lambda} \left( \frac{C_\delta L_\delta}{2\lambda} + C_{G*} \right) \triangleq C_\zeta; \tag{123}$$

*and if* $t \geq \tau_\alpha$, *then*

$$\mathbb{E}[\zeta(\theta_t, O_t)] \leq m_\zeta \alpha + m_\zeta' \tau_\alpha \alpha, \tag{124}$$

*where* $m_\zeta = 2C_\zeta$ *and* $m_\zeta' = C_G \left( \frac{L_J C_\delta L_\delta}{\lambda} + \frac{C_\delta L_\delta L_{G*}}{\lambda} + L_J C_{G*} \right)$.

Next we plug the tracking error (118) in (120).

$$\sum_{t=0}^{T-1} \frac{\alpha}{2} \mathbb{E}[||\nabla J(\theta_t)||^2]$$

$$\leq J(\theta_0) - J(\theta_T) + \alpha(\gamma + 2\gamma RK\varrho) \sqrt{\sum_{t=0}^{T-1} \mathbb{E}[||\nabla J(\theta_t)||^2]} \sqrt{2TQ_T + 2\sum_{t=0}^{T-1} M_t}$$

$$+ \alpha \tau_\alpha C_\zeta + \alpha^2 (T - \tau_\alpha)(m_\zeta + m_\zeta' \tau_\alpha) + \frac{L_J}{2} \alpha^2 C_G^2 T. \tag{125}$$

Divided both sides by $\frac{\alpha T}{2}$, we have that

$$\frac{\sum_{t=0}^{T-1} \mathbb{E}[||\nabla J(\theta_t)||^2]}{T}$$

$$\leq \frac{2J(\theta_0) - 2J(\theta_T)}{\alpha T} + 2(\gamma + 2\gamma RK\varrho) \sqrt{\frac{\sum_{t=0}^{T-1} \mathbb{E}[||\nabla J(\theta_t)||^2]}{T}} \sqrt{2Q_T + 2\frac{\sum_{t=0}^{T-1} M_t}{T}}$$

$$+ \frac{2\tau_\alpha C_\zeta}{T} + 2\alpha(m_\zeta + m_\zeta' \tau_\alpha) + L_J \alpha C_G^2. \tag{126}$$

We know from (118) that $2\frac{\sum_{t=0}^{T-1} M_t}{T} \le \frac{2}{1-e^{-u\beta}}\frac{3\alpha^2}{\lambda\beta}\frac{L_\delta^2}{\lambda^2}\frac{\sum_{t=0}^{T-1}\mathbb{E}[\|\nabla J(\theta_t)\|^2]}{T} + \frac{6}{\lambda\beta}\frac{4}{1-e^{-u\beta}}\|R_2\|^2$, thus

$$\frac{\sum_{t=0}^{T-1}\mathbb{E}[\|\nabla J(\theta_t)\|^2]}{T}$$

$$\le \frac{2J(\theta_0) - 2J(\theta_T)}{\alpha T} + 2(\gamma + 2\gamma RK\varrho)\sqrt{\frac{\sum_{t=0}^{T-1}\mathbb{E}[\|\nabla J(\theta_t)\|^2]}{T}}$$

$$\left(\sqrt{2Q_T + \frac{6}{\lambda\beta}\frac{4}{1-e^{-u\beta}}\|R_2\|^2} + \sqrt{\frac{2}{1-e^{-u\beta}}\frac{3\alpha^2}{\lambda\beta}\frac{L_\delta^2}{\lambda^2}\frac{\sum_{t=0}^{T-1}\mathbb{E}[\|\nabla J(\theta_t)\|^2]}{T}}\right)$$

$$+ \frac{2\tau_\alpha C_\zeta}{T} + 2\alpha(m_\zeta + m_\zeta'\tau_\alpha) + L_J\alpha C_G^2$$

$$= \frac{2J(\theta_0) - 2J(\theta_T)}{\alpha T} + 2(\gamma + 2\gamma RK\varrho)\sqrt{\frac{2}{1-e^{-u\beta}}\frac{3\alpha^2}{\lambda\beta}\frac{L_\delta^2}{\lambda^2}\frac{\sum_{t=0}^{T-1}\mathbb{E}[\|\nabla J(\theta_t)\|^2]}{T}}$$

$$\left(\sqrt{2Q_T + \frac{6}{\lambda\beta}\frac{4}{1-e^{-u\beta}}\|R_2\|^2}\right)2(\gamma + 2\gamma RK\varrho)\sqrt{\frac{\sum_{t=0}^{T-1}\mathbb{E}[\|\nabla J(\theta_t)\|^2]}{T}}$$

$$+ \frac{2\tau_\alpha C_\zeta}{T} + 2\alpha(m_\zeta + m_\zeta'\tau_\alpha) + L_J\alpha C_G^2$$

$$\triangleq \frac{2J(\theta_0) - 2J(\theta_T)}{\alpha T} + K_1\frac{\sum_{t=0}^{T-1}\mathbb{E}[\|\nabla J(\theta_t)\|^2]}{T} + K_2\sqrt{\frac{\sum_{t=0}^{T-1}\mathbb{E}[\|\nabla J(\theta_t)\|^2]}{T}} + \frac{2\tau_\alpha C_\zeta}{T}$$

$$+ 2\alpha(m_\zeta + m_\zeta'\tau_\alpha) + L_J\alpha C_G^2, \tag{127}$$

where $K_1 = 2(\gamma + 2\gamma RK\varrho)\sqrt{\frac{2}{1-e^{-u\beta}}\frac{3\alpha^2}{\lambda\beta}\frac{L_\delta^2}{\lambda^2}} = \mathcal{O}\left(\frac{\alpha}{\beta}\right)$ and $K_2 = \left(\sqrt{2Q_T + \frac{6}{\lambda\beta}\frac{4}{1-e^{-u\beta}}\|R_2\|^2}\right)2(\gamma + 2\gamma RK\varrho) = \mathcal{O}\left(\sqrt{\frac{\alpha^4}{\beta^2} + \frac{1}{T\beta} + \beta\tau_\beta}\right)$. Thus we can choose $\alpha$ and $\beta$ such that $K_1 \le \frac{1}{2}$, then we have that

$$\frac{\sum_{t=0}^{T-1}\mathbb{E}[\|\nabla J(\theta_t)\|^2]}{T}$$

$$\le \frac{4J(\theta_0) - 4J(\theta_T)}{\alpha T} + 2K_2\sqrt{\frac{\sum_{t=0}^{T-1}\mathbb{E}[\|\nabla J(\theta_t)\|^2]}{T}} + \frac{4\tau_\alpha C_\zeta}{T} + 4\alpha(m_\zeta + m_\zeta'\tau_\alpha) + 2L_J\alpha C_G^2$$

$$\triangleq U + V\sqrt{\frac{\sum_{t=0}^{T-1}\mathbb{E}[\|\nabla J(\theta_t)\|^2]}{T}}, \tag{128}$$

where $U = \frac{4J(\theta_0) - 4J(\theta_T)}{\alpha T} + \frac{4\tau_\alpha C_\zeta}{T} + 4\alpha(m_\zeta + m_\zeta'\tau_\alpha) + 2L_J\alpha C_G^2 = \mathcal{O}(\alpha\tau_\alpha + \frac{1}{\alpha T})$ and $V = 2K_2$. Hence, we have that

$$\frac{\sum_{t=0}^{T-1}\mathbb{E}[\|\nabla J(\theta_t)\|^2]}{T}$$

$$\le \left(\frac{V + \sqrt{V^2 + 4U}}{2}\right)^2$$

$$\overset{(a)}{\le} V^2 + 2U$$

$$\le 16\left(2Q_T + \frac{6}{\lambda\beta}\frac{4}{1-e^{-u\beta}}\|R_2\|^2\right)(\gamma + 2\gamma RK\varrho)^2 + \frac{8J(\theta_0) - 8J(\theta_T)}{\alpha T} + \frac{8\tau_\alpha C_\zeta}{T}$$

$$+ 8\alpha(m_\zeta + m_\zeta'\tau_\alpha) + 4L_J\alpha C_G^2$$

$$= \mathcal{O}\left(\frac{1}{T\alpha} + \alpha\tau_\alpha + \frac{1}{T\beta} + \beta\tau_\beta\right), \tag{129}$$

where $Q_T = \frac{\|z_0\|^2}{T(1-e^{-u\beta})} + \beta\frac{C_1}{u} + 4K^2\frac{\tau_\beta}{T} + \frac{c_z}{u(1-e^{-u\beta})T} + (m_z\beta + m_z'\tau_\beta\beta)\frac{1}{u}$.

### D.4 Constants

In this section we list all the constants occurred in our proof for the readers' reference.

$$C_\delta = c_{\max} + \gamma R \frac{\log |\mathcal{S}|}{\varrho} + (1+\gamma)K, \tag{130}$$

$$L_\delta = (1+\gamma), \tag{131}$$

$$L'_\delta = 2\gamma R\varrho, \tag{132}$$

$$L_J = 2\left(\frac{L_\delta^2}{\lambda} + \frac{C_\delta L'_\delta}{\lambda}\right), \tag{133}$$

$$C_1 = 3\left(C_\delta + \frac{C_\delta}{\lambda}\right)^2 + 3\left(C_\delta + (1+2R\varrho K)\frac{C_\delta}{\lambda}\right)^2, \tag{134}$$

$$C_G = C_\delta + \gamma K + 2\gamma\varrho R K^2, \tag{135}$$

$$C_H = C_\delta + K, \tag{136}$$

$$K_z = K + \frac{C_\delta}{\lambda}, \tag{137}$$

$$m_g = 4K_z\left(1 + \frac{1}{\lambda}\right)C_\delta, \tag{138}$$

$$m'_g = 4K_z L_\delta C_G\left(1 + \frac{1}{\lambda}\right) + C_\delta\left(1 + \frac{1}{\lambda}\right)\left(C_H + \frac{C_G C_\delta}{\lambda}\right), \tag{139}$$

$$m_f = 8K_z^2, \tag{140}$$

$$m'_f = 8K_z\left(C_H + \frac{C_G C_\delta}{\lambda}\right), \tag{141}$$

$$m_h = 2K_z C_h, \tag{142}$$

$$m'_h = C_h\left(C_H + \frac{C_\delta C_G}{\lambda}\right) + K_z L_h C_G, \tag{143}$$

$$C_{G*} = C_\delta + \frac{C_\delta}{\lambda}(\gamma + 2\varrho K\gamma R), \tag{144}$$

$$L_{G*} = L_\delta + \frac{L_\delta}{\lambda}(\gamma + 2\gamma R\varrho K) + \frac{C_\delta}{\lambda}L'_\delta, \tag{145}$$

$$L_h = \frac{L'_\delta}{L_\delta}C_h + \frac{L_\delta L_{G*}}{\lambda} + \frac{L_\delta L_J}{2\lambda}, \tag{146}$$

$$C_h = \frac{L_\delta}{\lambda}\left(C_\delta + \gamma(1-R) + 2\varrho K\gamma R\frac{C_\delta}{\lambda} + \frac{2L_\delta C_\delta}{\lambda}\right), \tag{147}$$

$$C_\zeta = \frac{C_\delta L_\delta}{\lambda}\left(\frac{C_\delta L_\delta}{2\lambda} + C_{G*}\right), \tag{148}$$

$$m_\zeta = 2C_\zeta, \tag{149}$$

$$m'_\zeta = C_G\left(\frac{L_J C_\delta L_\delta}{\lambda} + \frac{C_\delta L_\delta L_{G*}}{\lambda} + L_J C_{G*}\right) \tag{150}$$

## E  Experiments

**Experiments in Section 6.1:**

Frozen Lake Problem. We consider a $4 \times 4$ Frozen Lake problem. We set $\gamma = 0.96$, $\alpha = 0.8$.

Cart-Pole Problem. We set $\gamma = 0.95$, $\alpha = 0.2$.

**Experiments in Section 6.2:**

Frozen Lake Problem. We consider a $4 \times 4$ Frozen Lake problem. We set $\alpha = 0.1$, $\beta = 0.5$ and $\gamma = 0.9$. The initialization is $\theta = (1,1,1,1,1) \in \mathbb{R}^5$ and $\omega = (0,0,0,0,0)$. Each entry of every base function $\phi_s$ is generated uniformly at random between $(0,1)$.

**Additional Experiments on the Taxi Problem.**

We use the same setting as in Section 6.1 to demonstrate the robustness of our robust Q-learning algorithm. For the step size and discount factor, we set $\alpha = 0.3$ and $\gamma = 0.8$. The results are shown in fig. 5, from which the same observation that our robust Q-learning is robust to model uncertainty, and achieves a much higher reward when the mismatch between the training and test MDPs enlarges.

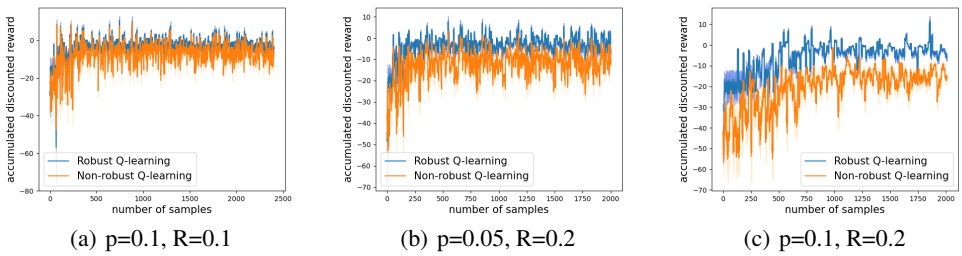

(a) p=0.1, R=0.1          (b) p=0.05, R=0.2          (c) p=0.1, R=0.2

Figure 5: **Taxi-v3**: robust Q-learning v.s. non-robust Q-learning.