# OpenReview forum: "Online Robust Reinforcement Learning with Model Uncertainty"
_NeurIPS.cc/2021/Conference — NeurIPS 2021 Poster_

### Official Review · Reviewer_VZkh · 2021-07-15

**Rating:** 7
**Confidence:** 3

**Summary:**

This paper focuses on model free robust Q-learning that can be implemented online. The authors estimate the unknown uncertainty set (R-contamination model) with current samples. Under the perfect duality condition they obtain the bellman operator which optimize their robust objectives. They propose two different algorithms: one for tabular case and another one with function approximation for problems with large state and action spaces. The robust algorithms presented in the paper converges as fast as their non-robust counterparts. They also provide empirical analysis to show the usefulness of their proposed methods.

**Limitations And Societal Impact:**

Yes.

**Main Review:**

The paper is well written and different ideas flow smoothly. The authors were able to make the content comprehensible.
(a)	Their model-free approach requires less space complexity compared to model based method while being robust to uncertainty.
(b)	They were able to prove that their Q-learning algorithm converges asymptotically to the optimal value function.
(c)	They prove that their algorithm converges as quickly as the vanilla Q-learning while being robust to uncertainty.
(d)	They provide a linear function approximation of their algorithm (Robust TDC) which can be used in domains with large state and action spaces.

The paper does clearly explain how they are different from the previous works which mostly address Non-robust model based RL algorithms. Their algorithm is robust to model uncertainty, model free and can be used online while maintaining the advantages of less space complexity and same convergence complexity as standard non-robust Q learning.

The authors only compare their robust Q-learning method against the standard Q-learning. It can be interesting to see how their algorithm perform when compared against other methods, for example different variants of policy gradient, and actor critic methods.

**Time Spent Reviewing:**

8

---

> ### Author Response · Authors · 2021-08-06
> **Response**
>
> We thank the reviewer for providing valuable feedback that helps us improve the quality of this paper. We also appreciate that the reviewer acknowledges our major technical novelty. Below is a point-to-point response to the questions and comments raised by the reviewer.
>
> Q1: It can be interesting to see how their algorithm performs when compared against other methods, for example different variants of policy gradient, and actor critic methods.
>
> A: There are a number of works on robust reinforcement learning through adversarial training and various algorithmic designs, which however do not come with theoretical robustness guarantees. We will include more numerical comparison with existing studies.

---

### Official Review · Reviewer_G5cv · 2021-07-16

**Rating:** 6
**Confidence:** 2

**Summary:**

This paper considers the model-free robust reinforcement learning, where the uncertainty set is assumed to be unknown. The authors propose a robust Q-learning algorithm for tabular settings and a robust TDC algorithm for function approximation settings. The authors theoretically analyze the converge properties for these two algorithms. Numerical experiments demonstrate the robustness of the proposed methods.

**Limitations And Societal Impact:**

Yes, the authors have adequately addressed the limitations and potential negative societal impact of their work.

**Main Review:**

This paper is theoretically sound. The authors present the analyses in detail, and the convergence guarantee is very nice. The algorithm and assumptions are natural in many applications. The paper is well-written. Moreover, the authors provide numerical experiments to show the effectiveness of the proposed methods. When considering the function approximation setting, the authors propose the TDC algorithm, which corresponds to the policy evaluation phase. It will be better if the authors could provide a brief discussion about the robust policy improvement.

**Time Spent Reviewing:**

6 hours.

---

> ### Author Response · Authors · 2021-08-06
> **response**
>
> We thank the reviewer for providing valuable feedback that helps us improve the quality of this paper. We also appreciate that the reviewer acknowledges our major technical novelty. Below is a point-to-point response to the questions and comments raised by the reviewer.
>
> Q1: It will be better if the authors could provide a brief discussion about the robust policy improvement.
>
> A: We appreciate the reviewer for pointing this out. Our approach can be readily extended to solve the optimal control problem with function approximation. For example, our approach can be generalized to robustify the Greedy Gradient Q-learning algorithm (Greedy-GQ) in [37]. Define the robust mean squared projected Bellman error similarly as $$J(\theta)\triangleq||\mathbf \Pi\mathbf TQ_{\theta}-Q_{\theta}||^2_{\mu},$$ where we replace the Bellman operator $\mathbf T_\pi$ in (12) with the robust Bellman operator $\mathbf T$ defined in line 146. Then our approach can be used to construct a two time-scale update rule to minimize $J(\theta)$. Note that the objective $J(\theta)$ is also non-convex, and our analysis for robust TDC can also be similarly extended.

---

### Official Review · Reviewer_Qzzj · 2021-07-31

**Rating:** 6
**Confidence:** 4

**Summary:**

The authors consider the robust reinforcement learning with uncertainty in the transition. A robust Q-learning algorithm is proposed for tabular MDPs and a robust TDC algorithm is proposed for compatibility of (linear) function approximation. Convergence rates are proved for both of the proposed algorithms. Empirical studies show that the two algorithms work for perturbed MDPs.

**Limitations And Societal Impact:**

See the main review.

**Main Review:**

The presentation (writing quality and clarity) is good until the section of robust TDC. The algorithm design is a solid contribution. Below are some major concerns.

1. The projection $\Pi$ should be presented with more clarity: Line 228--should include an explicit expression, which I believe would not take too much space. Line 248--since there is no explicit expression of $\Pi$, it is not completely clear from the context why the equation after "we know that" holds.

2. Two-timescale: While the convergence guarantee involving two stepsizes, the final simplified rate is obtained via setting the two timescales the same $O(1/T^{0.5})$. This makes me wonder: is two-timescale design/analysis really necessary?

3. Experiment and limitation: The setting for the experiments is $p\cdot \mathrm{uniform} + (1 - p)\cdot p_s^a$, which completely falls into the $R$-contamination uncertainty set. The results do show that the proposed algorithms work better than their vanilla counterparts. There are some cited works on robust RL. I think it would be better if the authors add comparisons with the methods proposed in the previous works in the experiment section. Moreover, even compared to the vanilla algorithms, do the robust versions always work better? What if (i) the MDP is not perturbed? (ii) the MDP is perturbed in a way that the $R$-contamination fails to cover? It would be nice if the authors can add discussion to such limitations and also add experiments to show what happens in those "unfavorable" cases.

Below are some minor comments:

4. The role of $K$: I feel that the radius $K$ plays an important role in robust TDC. From the view of reproducing kernel Hilber space, if $K$ is large enough, then with suitable feature, the model misspecification errors caused by the function approximation can be very small. I think the authors need to discuss this constant more. In addition, the choice of $K$ should also be mentioned in the experiment section. (This issue is less robustness related though)

5. Typo: Line 103 -- "provides", Line 246 -- definition of $C$ should be outer product. Clarity: Line 255 & 266/ Eq(17, 18) -- what kind of norm is used? Assumption 5 -- is it for all $\pi$?

**Time Spent Reviewing:**

6

---

> ### Author Response · Authors · 2021-08-07
> **Response**
>
> We thank the reviewer for providing valuable feedback that helps us improve the quality of this paper. We also appreciate that the reviewer acknowledges our major technical novelty. Below is a point-to-point response to the questions and comments raised by the reviewer.
>
> Q1: What is the exact definition of the $\mathbf{\Pi}$ and why does line 248 hold?
>
> A: We appreciate the reviewer for pointing this out. We will make this clear in the paper.
>
> Here, $\mathbf{\Pi}$ denotes the projection operator which projects a value function onto the function space $\mathcal{F}=${$V_{\theta}| \theta\in \mathbb{R}^N $} w.r.t. the norm $|| w||^2_{\mu_\pi}=w^\top D w$, where $D=diag(\mu_\pi(s_1), \mu_\pi(s_2),...,\mu_\pi(s_{|\mathcal{S}|}))$. More precisely, $ \mathbf{\Pi} v=V_{\theta}$ where $\theta= \arg\min_{\theta} ||v-V_{\theta}||^2_{\mu_{\pi}}$.
>
> For linear function approximation: $V_{\theta}=\Phi \theta$ where $\Phi=(\phi(s_1),...,\phi(s_{|\mathcal{S}|}))^\top\in\mathbb {R}^{|\mathcal S|\times N}$, the projection can be rewritten as  $\mathbf{\Pi} v=\Phi\cdot (\arg\min_{\theta} (v-\Phi\theta)^\top D(v-\Phi\theta))$. The minimization is over a quadratic function and the optimum is achieved at $\theta=(\Phi^\top D\Phi)^{-1}\Phi^\top D v$. Thus the projection operator $\mathbf{\Pi}$ can be written in a matrix form $\mathbf{\Pi}= \Phi (\Phi^\top D\Phi)^{-1}\Phi^\top D$. Line 248, i.e., $\mathbf{\Pi}^\top D\mathbf{\Pi}=D^\top\Phi(\Phi^\top D\Phi)^{-1}\Phi^\top D$ then follows from this linear expression of the projection operator.
>
>
> Q2: Why two time-scale is needed when the orders of both step sizes are the same?
>
> A: Although the two stepsizes have the same order, $\beta$ still has to be larger than $\alpha$ so that the tracking error is bounded within a desired range.
>
> We first explain why the two stepsizes are of the same order. Specifically, the upper bound of the tracking error is usually in the form of $K_1 \times Y$, where $K_1$ has the order of $\mathcal O(\frac{\alpha}{\beta})$. In many existing literatures of two time-scale algorithms, Y is usually derived as a constant, and then it is required that $\frac{\alpha}{\beta}\rightarrow 0$ so that the tracking error goes to zero. This usually leads to a convergence rate that is order-level worse than $\mathcal O(1/\sqrt{T})$. In our proof, we bound the tracking error as $K_1 \times \frac{\sum_{t=0}^{T-1}\mathbb E[||\nabla J(\theta_t)||^2]}{T}$, which is a more refined and tighter bound on the tracking error. We only require that $K_1$ to be less than a constant, e.g., 1/2, instead of $\rightarrow 0$ (see the exact requirement on $\alpha$ and $\beta$ in line 766). Note that $\frac{\sum_{t=0}^{T-1}\mathbb E[||\nabla J(\theta_t)||^2]}{T}$ is the gradient norm we would like to bound, which shall converge to zero. We exploit this idea and it leads to a convergence rate of $\mathcal O(1/\sqrt{T})$ in this paper.
>
> As discussed above, although the two stepsizes are of the same order, we still need one step size $\beta$ to be larger than the other step size $\alpha$ so that the tracking error is under control. In other words, if we only use one time scale, i.e., letting $\alpha=\beta$, then the tracking error will be too large, and the algorithm may not converge.
>
> Q3 part 1: it would be better if the authors add comparisons with the methods proposed in the previous works in the experiment section.
>
> A: There are a number of works on robust reinforcement learning through adversarial training and various algorithmic designs, which however do not come with theoretical robustness guarantees. Despite of that, we agree with the reviewer that comparisons with these methods will still be insightful. We will include more numerical comparison with existing studies.
>
> Q3 part 2: Do robust algorithms always outperform the vanilla ones?
> A: We agree with the reviewer that the robust algorithm may not always outperform the vanilla one. We will include more examples to comprehensively demonstrate the performance of our robust algorithms under various settings.
>
> For example, when the MDP is not perturbed at all, the vanilla algorithm will outperform our robust algorithm. However, we would like to highlight that the goal of the robust algorithm is to guarantee a "good” enough performance no matter which MDP in the uncertainty set is the true one, and is not to achieve the optimal performance for every MDP in the uncertainty set. We note that the latter one is not possible since only samples from the "centroid" MDP is available.
>
> Regarding the case where the MDP is perturbed in a way that the R-contamination model fails to cover, e.g., p is larger than $R$, it is not clear how the algorithm will perform if the true model falls outside of the uncertainty set. In practice, the uncertainty set shall be “large” enough to contain all or most distributions of interest, yet “small” enough to avoid being overly conservative. If $R$ is too small, no robustness can be guaranteed; and if $R$ is chosen too large, the worst-case is too pessimistic to be useful. We will include some experiments to illustrate this scenario in the paper.
>
> Minor comments:
>
> Q4: Role of K.
>
> A: We agree with the reviewer that with a large enough $K$ and suitable feature, the model misspecification error shall be very small. We will also specify the $K$ used in the experiments in the paper.
>
> Q5: Typo
>
> A: We appreciate the reviewer for pointing out the typos. We have corrected these typos. The norm used in line 255 and line 266 is L2-norm. Assumption 5 is only for the behavior policy $\pi$, not all the policies. We will make these clear in the paper.

---

### Official Review · Reviewer_c2hD · 2021-07-31

**Rating:** 7
**Confidence:** 4

**Summary:**

I thank the authors for their responses. They main point of remaining concern seems to be the necessity of analysis/evaluation under model misspecification. I feel like this concern is quite significant, but would defer to better experts in the field.

=====

The paper presents robust versions of the Q-learning and TDC algorithms and analyzes their rate of convergence in tabular and linear-approximation representations.

**Limitations And Societal Impact:**

The authors adequately address the limitations of their work.

**Main Review:**

There's one central point of high significance and (to my knowledge) originality in this paper, which is the idea of thinking of each individual data point as a degenerate model, and showing that being robust w.r.t. each point averages out to being robust to the average. The analysis itself is incremental (i.e. doesn't introduce new tools) but strengthens the result with guarantees — it's also interesting that they nearly match guarantees without robustness. The paper is also written very clearly.

As a non-expert in robust RL, I was missing some justification of the setting. An adversary that can perturb each step independently and with knowledge of past agent actions seems very strong, and robustness to it may give an overly cautious agent. It's also unclear to me why a uncertainty set of the form (4) is reasonable (e.g. where would it be applicable? how does it compare with KL level sets?), and in what sense the true dynamics is its “centroid”.

It makes sense to leave proofs for an appendix, but this should be indicated clearly. The authors should also consider providing some intuitive overview of the proofs, particularly involving the novel aspects. For example, a discussion of how (6) averages out to provide convergence can be insightful.

The experiments are very limited, including only two very simple domains, with 6.2 even excluding one of them (why?). How was R selected in each experiment?

**Time Spent Reviewing:**

4

---

> ### Author Response · Authors · 2021-08-07
> **response**
>
> We thank the reviewer for providing valuable feedback that helps us improve the quality of this paper. We also appreciate that the reviewer acknowledges our major technical novelty. Below is a point-to-point response to the questions and comments raised by the reviewer.
>
> Q1: An adversary that can perturb each step independently and with knowledge of past agent actions seems very strong, and robustness to it may give an overly cautious agent.
>
> A: We do not assume that the adversary knows past actions of the agent. The uncertainty set can be interpreted as that the adversary randomly and independently perturbs the transition with probability $R$ in an arbitrary way. The goal is to optimize the worst-case value function over such an uncertainty set. As also will be discussed later in detail, the $R$-contamination uncertainty set is a subset of a total variation defined uncertainty set, and thus is also related to a KL divergence defined uncertainty set via Pinsker’s inequality and many other divergence metrics, e.g., Hellinger distance.
>
> The dynamic model, which allows the transition kernel $\mathsf P_t$ (i.e., the adversary strategy) to be time-varying, is actually equivalent to the static model, where the transition kernel $\mathsf P_t$ is static, i.e., $\mathsf P_{t_1}=\mathsf P_{t_2}$ for any $t_1$ and $t_2$, under a general condition that the agent’s policy is stationary and the problem is infinite horizon with discounted reward [25]. Therefore, the optimal robust policy under the dynamic model and under the static model is the same, and the dynamic model does not make the adversary overly strong.
>
> Q2: Why are the R-contamination uncertainty sets reasonable and where could it be applied?
>
> A: The $R$-contamination model dates back to the seminal work of Huber on robust statistics in the 1960s [24], and is a widely used model for distributional uncertainty. From its definition, the $R$-contamination uncertainty set models the scenario where the MDP state transition is perturbed independently and randomly in an arbitrary way with a small probability $R$, and thus it is more suitable for random and arbitrary attacks in adversarial environments, outliers in data samples, and unknown and unexpected system perturbation. For example, a drone flying in the air may be pushed by some unexpected air-flow from any direction and with any strength.
>
> Moreover, the $R$-contamination model is a subset of a total variation defined uncertainty set with radius $R$, and thus is also related to uncertainty sets defined by many other distance measures, e.g., KL divergence, Hellinger distance. Our results for the $R$-contamination model also suggests robustness under uncertainty sets defined by many other distance metrics.
>
> Q3: How do R-contamination sets compare with other sets like KL level sets?
>
> A: The $R$-contamination uncertainty set is actually closely related to the KL divergence defined uncertainty set.
>
> It can be shown that the $R$-contamination uncertainty set centered at $p$ is a subset of the total variation defined uncertainty set centered at $p$: {$(1-R)p+Rq: q\in \Delta^{|S|}$}$\subseteq${$ q: d_{TV}(q,p)\leq R$}. Hence total variation defined uncertainty set (with radius $R$) is less conservative than our $R$-contamination uncertainty set.
>
> On the other hand, total variation is closely related to KL divergence. Both of them are instances of a more general family called $f$-divergence: $d_f(p,q)=\int_{\Omega} f \left( \frac{dp}{dq}\right) dq$.
>
> Pinsker’s inequality characterize their relation: $d_{TV}(p,q)\leq \sqrt{\frac{1}{2} d_{KL}(p \| q)}$. This inequality implies that a KL divergence defined uncertainty set with radius $R$ is a subset of a total variation defined uncertainty ball with radius $\sqrt{R/2}$. However, there is no direct relationship between the $R$-contamination uncertainty set and the KL divergence defined uncertainty set. It is also of our future interest to develop model-free robust RL method for KL-divergence defined uncertainty set.
>
>
> Q4: In what sense the true dynamics is its “centroid”.
>
> A: Thanks for the suggestion. It may be more appropriate to call it the “idealized model” following the notion of Huber in [24] instead of “centroid”.
>
> Q5: It makes sense to leave proofs for an appendix, but this should be indicated clearly. The authors should also consider providing some intuitive overview of the proofs, particularly involving the novel aspects. For example, a discussion of how (6) averages out to provide convergence can be insightful.
>
> A: We thank the reviewer for the suggestion, and we will provide clear reference in the main paper to the appendix for the proof. We will also include discussion on the insights behind the proof in the paper.
>
> We agree with the reviewer that a discussion of how (6) averages out to provide convergence will be insightful. We will include the following discussion in the paper. According to the update rule in (6), conditioning on $(s_t,a_t)$, the expectation of the right hand side of (6) is equivalent to the one from applying the robust Bellman operator to $Q_t$. Note that the robust Bellman operator is a contraction with the fixed point being the optimal robust $Q^*$. Therefore, repeatedly applying (6), the algorithm converges to $Q^*$.
>
> Q6: The experiment part is limited with only two simple domains.
>
> A: We also have some additional experiments in the appendix due to the space limitation. We will perform more experiments to demonstrate the proposed approach comprehensively.
>
> Q7:  How is R selected in the experiments?
>
> A: In our experiments, we choose $R$ to be larger than $p$ so that the perturbed model is within the uncertainty set. In practice, the choice of R corresponds to the size of the uncertainty set. It shall be “large” enough to contain all or most distributions of interest, yet “small” enough to avoid being overly conservative. If $R$ is too small, no robustness can be guaranteed; and if $R$ is chosen too large, the worst-case is too pessimistic to be useful.

---

> > ### Comment · Reviewer_c2hD · 2021-08-18
> > **Further Discussion**
> >
> > During the discussion, a concern was raised that the paper is missing a theoretical analysis or empirical evaluation of the behavior of the proposed method under a misspecified perturbation model. This seems important, because there's no a-priori way to correctly specify the perturbation model in a given instance. Would the authors please clarify their view of this concern?

---

> > > ### Author Response · Authors · 2021-08-19
> > > **Further Response**
> > >
> > > First of all, investigating how the algorithm performs is of little usage when the $R$-contamination model does not fit, and we do not encourage the use of our algorithm in this case. It is of more importance to design a suitable uncertainty set, and further design online and model-free robust RL algorithms. Our robust algorithms are guaranteed to work well when the true MDP falls into the $R$-contamination uncertainty set, e.g., flying drones in extreme environments with arbitrary and unpredictable airflow.
> > >
> > > Secondly, for robust RL or more generally distributionally robust learning and inference, where there exists uncertainty about the underlying data generating distribution (see some example references listed below and also references [25, 42, 5, 49, 62, 41, 48] in the paper), to the best of the authors’ knowledge, the theoretical investigation into the case with misspecified uncertainty set and the case when the test model falls out of the uncertainty set is rare. Based on the authors’ understanding, this is mainly due to its little practical usage as it is more important to design a suitable uncertainty set and associated algorithms. We could demonstrate several examples empirically in the paper, but we do not expect any useful insights from doing that.
> > >
> > > Thirdly, our $R$-contamination model is quite general and is a widely used model in practice to model adversarial, random and arbitrary perturbation. When $R=1$, the $R$-contamination uncertainty set includes any possible MDPs. When $R=0$, our model reduces to a singleton at the training MDP. The size of the uncertainty set changes with $R$. As discussed in our previous response, the $R$-contamination model covers a wide range of applications and can be easily connected with various uncertainty sets defined using other distances, e.g., KL divergence, total variation. Despite its generality, we do not aim to cover all possible MDPs using the $R$-contamination model, and investigation into uncertainty sets defined by other distance metrics is of our future interest. In practice, choosing a proper uncertainty set and setting a suitable radius are done based on domain knowledge and expert design. For example, to fly a drone in extreme weather, it is to the practitioner’s choice how bad weather he/she would like the drone to handle, and the radius $R$ can be chosen based on how often the drone is hit by unexpected airflow. The minimax setting in this paper guarantees a good enough performance under all possible scenarios in the uncertainty set, however, the price needs to be paid is the performance loss when the weather is not actually that bad.
> > >
> > >
> > >
> > >
> > >
> > > [*1] Gao, Rui, Xi Chen, and Anton J. Kleywegt. "Wasserstein distributionally robust optimization and variation regularization." arXiv preprint arXiv:1712.06050 (2017).
> > >
> > > [*2] Staib, Matthew, and Stefanie Jegelka. "Distributionally robust optimization and generalization in kernel methods." Advances in Neural Information Processing Systems 32 (2019): 9134-9144.
> > >
> > > [*3] Zhu, Jia-Jie, et al. "Kernel Distributionally Robust Optimization: Generalized Duality Theorem and Stochastic Approximation." International Conference on Artificial Intelligence and Statistics. PMLR, 2021.
> > >
> > > [*4]Gül, Gökhan, and Abdelhak M. Zoubir. "Minimax robust hypothesis testing." IEEE Transactions on Information Theory 63.9 (2017): 5572-5587.
> > >
> > > [*5] Rahimian, Hamed, and Sanjay Mehrotra. "Distributionally robust optimization: A review." arXiv preprint arXiv:1908.05659 (2019).
> > >
> > > [*6] Fauß, Michael, Abdelhak M. Zoubir, and H. Vincent Poor. "Minimax Robust Detection: Classic Results and Recent Advances." IEEE Transactions on Signal Processing 69 (2021): 2252-2283.

---

### Decision · Program_Chairs · 2021-09-27

**Decision:**

Accept (Poster)

**Comment:**

Based on the reviews and the discussion afterwards, my recommendation weighs towards acceptance. While the reviewers reached a consensus that the algorithms make novel and solid contributions, there remain some criticisms. First, the experiments can be made much more solid and comprehensive by adding previous robust RL methods as baselines, showing the tradeoff between robustness and pure performance, and presenting the performance evaluation in the face of misspecified uncertainty set. Second, I encourage the authors to also add discussion (and possibly theories) to address the limitation of the R-contamination model. I hope that the above concerns, along with other clarity issues raised by the reviewers, are addressed in the camera-ready version.